# Hypersensitivity of glacial summer temperatures in Siberia

Pepijn Bakker[1,2], Irina Rogozhina[3,2,4], Ute Merkel[2], and Matthias Prange[2]

[1]Department of Earth Sciences, Vrije Universiteit Amsterdam, Amsterdam, The Netherlands
[2]MARUM-Center for Marine Environmental Sciences, University of Bremen, Bremen, Germany
[3]Norwegian University of Science and Technology, Trondheim, Norway
[4]Institute of Physics of the Earth, Russian Academy of Science, Russia

**Correspondence:** Pepijn Bakker (p.bakker@vu.nl)

**Abstract.** Climate change in Siberia is currently receiving a lot of attention because large permafrost-covered areas could provide a strong positive feedback to global warming through the release of carbon that has been sequestered there on glacial-interglacial time scales. Geological evidence and climate model experiments show that the Siberian region also played an exceptional role during glacial periods. The region that is currently known for its harsh cold climate did not experience major glaciations during the last ice age, including its severest stages around the Last Glacial Maximum (LGM). On the contrary, it is thought that glacial summer temperatures were comparable to present-day. However, evidence of glaciation has been found for several older glacial periods.

We combine LGM experiments from the second and third phases of the Paleoclimate Modelling Intercomparison Project (PMIP2 and PMIP3) with sensitivity experiments using the Community Earth System Model (CESM). Together these climate model experiments reveal that the intermodel spread in LGM summer temperatures in Siberia is much larger than in any other region of the globe and suggest that temperatures in Siberia are highly susceptible to changes in the imposed glacial boundary conditions, the included feedbacks and processes, and to the model physics of the different components of the climate model. We find that changes in the circumpolar atmospheric stationary wave pattern and associated northward heat transport drive strong local snow and vegetation feedbacks and that this combination explains the susceptibility of LGM summer temperatures in Siberia. This suggests that a small difference between two glacial periods in terms of climate, ice buildup or their respective evolution towards maximum glacial conditions, can lead to strongly divergent summer temperatures in Siberia, allowing for the buildup of an ice sheet during some glacial periods, while during others, above-freezing summer temperatures preclude a multi-year snow-pack from forming.

## 1 Introduction

During the Last Glacial Maximum (LGM; ~24-18 ka) ice sheets covered large parts of the Northern Hemisphere continents. Over North America and northwestern Eurasia continental ice sheets extended from the Arctic ocean down to ~40°N in some areas. A notable exception was northeastern Siberia, a region that remained largely ice-free during the LGM according to archaeological evidence (Pitulko et al., 2004), geological reconstructions and permafrost records (Boucsein et al., 2002; Schirrmeister, 2002; Hubberten et al., 2004; Gualtieri et al., 2005; Stauch and Gualtieri, 2008; Wetterich et al., 2011; Jakobsson

et al., 2014; Ehlers et al., 2018), and combined model-data driven ice-sheet reconstructions (Abe-Ouchi et al., 2013; Kleman et al., 2013; Peltier et al., 2015). This is intriguing given the fact that the area presently extends as far north as ~75°N, and extended even further north during the LGM when a large part of the Siberian continental shelf was exposed because of eustatic sea-level lowering.

Reconstructing Quaternary ice sheet limits and assigning geological ages has for various reasons proven a difficult task for the Siberian region (e.g., Jakobsson et al., 2014). Svendsen et al. (2004) synthesized the existing geological data and concluded that since the penultimate glacial period (~140 ka), most of Arctic Siberia has remained ice-free, with the exception of the high-altitude Putorana Plateau and the coastal areas of the Kara Sea. Independent evidence from permafrost records (Boucsein et al., 2002; Schirrmeister, 2002; Hubberten et al., 2004; Wetterich et al., 2011), marine sediment cores (Darby et al., 2006;

Polyak et al., 2004, 2007, 2009; Adler et al., 2009; Backman et al., 2009) and dating of mollusc shells (Basilyan et al., 2010) also indicates that the entire region between the Taymyr Peninsula and the Chukchi Sea remained ice free and was covered by tundra-steppe during the LGM and that the last grounded ice impacts in different sectors of this region are dating back to MIS 6, or potentially MIS 5 within the dating uncertainties (Stauch and Gualtieri, 2008). Hence, the existing geological evidence indicates that ice sheets covered large parts of western Siberia (Svendsen et al., 2004; Patton et al., 2015; Ehlers et al., 2018)

and the East Siberian continental shelf (Niessen et al., 2013; Jakobsson et al., 2014, 2016) prior to the last glacial period but it remains unclear how often northeastern Siberia experienced large-scale glaciations during the different glacial periods of the Quaternary. Nonetheless, it appears that this far northern region was covered by ice during some glacial periods, while it remained ice free during others.

  A number of studies have simulated the East Siberian LGM climate and ice sheet growth (e.g. Krinner et al., 2006; Charbit

et al., 2007; Ganopolski et al., 2010; Abe-Ouchi et al., 2013; Beghin et al., 2014; Peltier et al., 2015; Liakka et al., 2016). They show widely different results, from ice-free conditions to the buildup of a large ice sheet covering most of Siberia, and therefore the correspondence with proxy-based reconstructions ranges from good to very poor.

  Over the years, a number of possible mechanisms have been suggested to explain the lack of an ice sheet covering eastern Siberia during the LGM, and perhaps therewith also explain the divergent results of coupled climate-ice-sheet simulations for

this region during the LGM. The most widely discussed mechanisms involve changes in atmospheric dust load, orographic precipitation effects and/or changes in atmospheric circulation driven by the buildup of the North American and/or Eurasian Ice Sheets.

  During glacial times, the atmospheric dust load and dust deposition was likely substantially larger, particularly at the southern margins of the Northern Hemisphere ice sheets and over Siberia (Harrison et al., 2001; Lambert et al., 2015; Mahowald et al.,

1999, 2006). Modelling studies have shown that the buildup of ice over Siberia can be strongly impacted by the effect of dust on the surface albedo as an increase of dust deposition on the snow pack leads to a lowering of the snow albedo that in turn leads to higher melt rates (Krinner et al., 2006; Willeit and Ganopolski, 2018).

  Continental ice sheets have a strong impact on the climate. It was already recognized by Sanberg and Oerlemans (1983) that under the influence of a preferred wind direction, an ice sheet can create a distinct asymmetry with high precipitation rates at

the windward side and low precipitation rates on the leeward side. This precipitation shadow effect has also been proposed

as an explanation for a westward migration of the Eurasian Ice Sheets during the last glacial period (Liakka et al., 2016, and references therein). Through the precipitation shadow effect, the buildup of the Eurasian Ice Sheet would lead to dry conditions in Siberia and potentially prevent the buildup of an ice sheet in the area.

Another way how ice sheets can impact the climate is through their steering effect on the large-scale atmospheric circulation. Broccoli and Manabe (1987) showed that the buildup of the North American ice sheets leads to substantial changes in the mid-tropospheric flow, including a split of the jet-stream around the northern and southern edges of the ice sheet and a resulting increase of summer temperatures over Alaska. Similar impacts of glacial ice sheets on large-scale atmospheric circulation were found in a number of other modelling studies (e.g. Cook and Held, 1988; Roe and Lindzen, 2001; Justino et al., 2006; Abe-Ouchi et al., 2007; Langen and Vinther, 2009; Liakka and Nilsson, 2010; Ullman et al., 2014; Liakka et al., 2016). Generally these studies indicate a warming over Alaska as a result of the growth of the North American ice sheets, but it differs from one study to the next how far westward this warming extends into Siberia. In these modelling studies the warming in Alaska and Siberia is linked to increased poleward heat transport induced by changes in the atmospheric stationary waves and to local feedbacks involving the surface albedo and atmospheric water vapor content (Liakka and Lofverstrom, 2018). A compilation of LGM temperature reconstructions based on various land proxy data provides support to these inferences, showing that LGM summer temperatures in Northern Siberia were overall not very different from the relatively mild present-day summer temperatures in the region (Meyer et al., 2017).

The lack of an LGM ice cover in northeastern Siberia has often been attributed to the increased atmospheric dust load and/or a precipitation shadow effect of the Eurasian Ice Sheet to the west. However, based on these mechanisms alone one cannot readily explain the absence of a Siberian ice sheet in some glacial periods, but its presence in others, or reconstructions of Siberian LGM summer temperatures close to present-day values (Meyer et al., 2017), suggesting that these processes are likely only part of the story. Existing and new coupled climate model results can shed light on these intriguing geological observations. Here we show that the inter-model spread of simulated LGM summer temperatures is exceptionally large in Siberia compared to any other region, suggesting a high susceptibility of Siberian summer temperatures to minor changes in boundary conditions or model formulation, and discuss potential underlying mechanisms and causes. We argue that this high susceptibility of Siberian summer temperatures to boundary conditions (hypersensitivity) is a major factor for the absence or presence of ice sheets in different Quaternary glacials.

## 2   Methodology

In this study we combine LGM simulations from the second and third phases of the Paleoclimate Modelling Intercomparison Project (PMIP2 and PMIP3) with LGM sensitivity experiments using the Community Earth System Model (CESM).

### 2.1   PMIP experiments

We use 17 LGM coupled climate model simulations from PMIP2 and PMIP3/CMIP5 (Table 1; Braconnot et al., 2007; Harrison et al., 2015) and their corresponding pre-industrial (PI) control simulations as a reference. LGM boundary conditions follow the

PMIP2 and PMIP3 protocols and include reduced greenhouse-gas concentrations, changed astronomical parameters, prescribed continental ice sheets and a lower global sea level. Nearly half (7/17) of these simulations include dynamic vegetation while

the remainder uses prescribed PI vegetation (Table 1). See https://pmip2.lsce.ipsl.fr and https://pmip3.lsce.ipsl.fr for further details and references. The analysis of PMIP model output is based on climatological means and all output was regridded to a common $0.9°x1.25°$ horizontal resolution. In order to compare the sea-level pressure results from different models and between PI and LGM we removed the respective global mean before calculating the anomalies. For the analysis of geopotential height fields only 16 instead of 17 PMIP models are used because PMIP2 LGM geopotential height from ECHAM5-MPIOM

was not available to us.

## 2.2   CESM experiments

To study the simulated LGM temperatures in the Siberian region in more detail and to isolate individual mechanisms, we analyzed a number of sensitivity experiments performed with the state-of-the-science coupled climate model CESM (version 1.2; Hurrell et al., 2013). The model includes the Community Atmosphere Model (CAM), Community Land Model (CLM4.0),

the Parallel Ocean Program (POP2) and the Community Ice Code (CICE4). In all our CESM experiments, we use a horizontal resolution of $1.9°x2.5°$ in the atmosphere (finite volume core) and land, and a nominal $1°$ resolution of the ocean (60 levels in the vertical) and sea-ice models with a displaced North Pole.

For the CESM LGM simulations we followed the most recent PMIP protocol (PMIP4; Kageyama et al., 2017), including greenhouse-gas concentrations (190 ppm $CO_2$, 357 ppb $CH_4$ and 200 ppb $N_2O$), orbital parameters (eccentricity of 0.019,

obliquity of $22.949°$ and perihelion-$180°$ of $114.42°$) and changes in the land-sea distribution and altitude due to lower sea-level (Di Nezio et al., 2016). In this study we used as default the GLAC-1D LGM ice sheet reconstruction (Ivanovic et al., 2016). Note that the PMIP4 $CH_4$ concentration of 375 ppb is slightly higher than the one used here.

In the first set of sensitivity experiments we altered the imposed LGM ice sheet boundary conditions. Within the framework of PMIP4 two LGM ice sheet reconstructions are suggested as boundary conditions for the LGM experiments (Kageyama

et al., 2017), namely GLAC-1D (Ivanovic et al., 2016) and ICE-6G (Peltier et al., 2015). When comparing these two ice sheet reconstructions we find substantial differences, especially an overall increase of the height of the North American ice sheets in ICE-6G compared to GLAC-1D and a lowering of the Eurasian Ice Sheet (Figure 5A; both differences are on the order of 10% of the total ice sheet height, for more details see Kageyama et al., 2017). Changes in surface roughness resulting from the ice sheet changes are highly uncertain and have not been taken into account. We performed a set of experiments to investigate the

impact of these two different ice sheet reconstructions on simulated Siberian LGM temperatures (see "Continental ice sheets" set of experiments in table 2 ).

In the second set of sensitivity experiments, we used two different versions of the atmosphere model, CAM4 and CAM5, to investigate the importance of the atmospheric model physics (see "Atmospheric model physics" set of experiments in table 2). CAM5 differs from its predecessor because it simulates indirect aerosol radiative effects by including full aerosol-cloud interactions. Furthermore, it includes improved schemes for moist turbulence, shallow convection and cloud micro- and macro-physics. Finally, while CAM4's grid has 26 vertical levels, in CAM5 four levels were added near the surface for a better

representation of boundary layer processes. See Neale et. al. (2010) for a more detailed description of the atmospheric models used in CESM.

Furthermore, the land model CLM4.0 includes the possibility to use a representation of the carbon-nitrogen cycle and to calculate the resulting changes in leaf area index, stem area index and vegetation heights per plant-functional-type (Lawrence et al., 2011). These changes in the biophysical properties of the vegetation cover impact, for instance, evapotranspiration and surface albedo. Note that the spatial distribution of plant-functional-types is prescribed in CLM4.0, which is why the model is sometimes described as a semi-dynamic vegetation model. Nonetheless, for simplicity we will refer to simulations that

include carbon-nitrogen dynamics as 'interactive vegetation' simulations in the remainder of this manuscript. To study the interdependency of interactive vegetation and atmospheric model physics we performed a total of four experiments with either CAM4 or CAM5 and including or excluding interactive vegetation that are referred to as the "Interactive vegetation" set of experiments (Table 2).

All LGM experiments performed with CESM start from a previous LGM simulation and are run for at least 200 years to

obtain a new surface climate equilibrium. Carbon pools in the litter and soils take centuries to equilibrate. However, we find that the trends are sufficiently small after 200 years to perform a robust analysis of the surface climate. Changes in Siberian (global) vegetation carbon pools amount to less than 2% (0.6%) of the total PI-to-LGM change for the model years 150-200. Top-of-the-atmosphere imbalances in the simulations including the carbon-nitrogen cycle are -0.1$\mathrm{Wm}^{-2}$ and -0.185$\mathrm{Wm}^{-2}$ using the CAM4 and CAM5 atmospheric models, respectively. Climatologies are calculated based on the last 30 years of

the simulations. For the sensitivity experiments focusing on interactive vegetation and atmospheric model physics, we also performed corresponding PI simulations (Table 2) to enable a proper analysis. Our five CESM LGM experiments are jointly referred to as the CESM LGM ensemble.

Throughout this manuscript we focus on boreal summer (June-July-August; JJA) near-surface air temperatures and simply referred to it as 'JJA temperatures' in the remainder of this manuscript. Moreover, when calculating LGM anomalies, we refer

to the difference between an LGM simulation and the corresponding PMIP or CESM PI experiment (Table 2). It is in turn differences between these CESM LGM anomalies that we use to highlight mechanisms behind the susceptibility of Siberian summer temperatures (Section 3.2).

## 3  Results

### 3.1  Siberian LGM temperatures in PMIP2 and PMIP3 ensemble

The combined PMIP2 and PMIP3 LGM experiments reveal the particularity of LGM JJA temperatures in Siberia. Of all continental areas that were not covered by large ice sheets, Siberia shows the largest inter-model spread of LGM anomalies (standard deviation; Figure 1B). Another striking feature of the Siberian region is that it is one of the few regions where the PMIP multi-model mean temperature anomaly is close to, or even above zero in some areas, indicating that LGM summers were potentially as warm as at present (Figure 1A). Taken together, PMIP simulations show LGM JJA temperatures in Siberia ranging from warmer to substantially colder than at present. If we define a target region for Siberia based on the area where the

PMIP multi-model spread is larger than 7°C (green contours in Figure 1; referred to as "Siberian target region" in the remainder of the manuscript and located roughly between 120°E-180°E and 70°N-75°N), we see that JJA temperature anomalies averaged over the target region for the individual models range between -12°C and +12°C (Figure 2 and Tabe 1). The spread in simulated LGM temperatures in the Siberian target region increases compared to PI in all seasons, however JJA really stands out (top row figure A1).

Disentangling the causes of the particularity of the Siberian LGM summer temperatures based on PMIP results isn't straightforward because of multiple possible underlying causes; nonetheless, some aspects can be identified. Whereas the simulated temperature changes are quite different among PMIP models, a robust decrease in precipitation on the order of 20-30% is simulated (Figure 1C and 1D). As a consequence, the (Pearson) correlation between temperature change and precipitation change in the target region is insignificant at the 0.05 significance level (R=0.36; Figure 2A; note that throughout the manuscript, correlation refers to inter-model correlation). A significant correlation is found between temperature and snow cover, with higher temperatures corresponding to a lower snow cover (R=-0.60; p<0.05; Figure 2E). There are similarities between the spatial patterns of the PMIP multi-model spread in temperature anomalies and cloud cover anomalies (Figure 1B and 1F), however, within the Siberian target region local JJA temperature anomalies and cloud cover anomalies are not correlated at the 0.05 significance level (R=-0.45; Figure 2B), arguing against a leading role of local cloud dynamics to explain the large inter-model spread in Siberian temperatures. As in Yanase and Abe-Ouchi (2007), we find that a weakening of the North Pacific high during JJA is a consistent feature of PMIP LGM simulations (Figure 1G). Moreover, a strong anticorrelation is found in the PMIP LGM simulations between JJA temperature and sea-level pressure anomalies over the Siberian target region (R=-0.72; p<0.05; Figure 2C): a more positive temperature anomaly locally creates a thermal low and hence corresponds to a less pronounced sea-level pressure anomaly. Concurrently, higher sea-level pressure anomalies correspond to more positive cloud cover anomalies (R=0.50; p<0.05; Figure 2D). Liakka et al. (2016) found in their model that higher pressure is associated with lower cloud cover that in turn leads to an increase in JJA temperatures, but our results suggest that this is not the leading mechanism in the majority of PMIP LGM results. Inspecting the PI and LGM seasonal cycles for cloud and snow cover, we find that also for these variables the changes in inter-model spread in Siberia are most pronounced in summer. In contrast, the inter-model spread in precipitation doesn't change much between PI and LGM (Figure A1).

The strong negative correlation between JJA temperature and sea-level pressure anomalies suggests that the sea-level pressure changes could be a consequence of local temperature changes. Indeed, another reason for the negative correlation could be a remote forcing through anomalous heat advection into the Siberian target region. We find evidence for such a remote forcing of the temperature variations in the Siberian target region in the significant correlation with the large-scale mid-to-high latitude stationary wave pattern, resembling a wavenumber 2 structure (Figure 3). Increased Siberian JJA temperatures correspond with a lowering (increasing) of the JJA 500 hPa geopotential height to the southwest (southeast) of the region. The remote forcing of Siberian temperatures can thus be the result of an increase in northward flowing relatively warm air masses over the eastern part of the Asian continent into the region of interest.

A deeper understanding of the large multi-model spread in PMIP LGM JJA temperatures over Siberia and of the mechanisms proposed above is hampered by a multitude of differences between PMIP simulations: different model formulations, different

parts of the climate system that are included and different boundary conditions including the uncertainty in the reconstructed
LGM ice sheet and continental outlines. Moreover, certain key climate variables are not available for a sufficiently large number
of the PMIP models. In the following we will therefore investigate a purpose-built CESM-based ensemble of LGM simulations
with clearly defined differences between the individual sets of sensitivity experiments.

## 3.2   Siberian LGM temperatures in CESM ensemble

We construct three sets of LGM sensitivity experiments performed with the CESM climate model in order to investigate in
more detail the impact of changes in boundary conditions (continental ice sheets), model formulations (atmospheric model
physics) and including different components of the climate system (interactive vegetation; Table 2).

Despite the fact that our total CESM LGM ensemble is smaller than the PMIP ensemble (n=5 instead of n=17) and that it
wasn't designed to mimic the PMIP ensemble, we find that the spread in the CESM LGM temperature anomalies is surprisingly
similar to the PMIP multi-model spread, both in terms of spatial distribution as well as magnitude (Figure 4B). This gives us
confidence that investigating the causes of the sensitivity of northeastern Siberian temperatures in the CESM ensemble can
provide insights into the PMIP inter-model differences. JJA temperatures in the Siberian target region for the individual CESM
experiments are listed in table 3.

First we analyze the first set of experiments ("Continental ice sheets"), differing only in the imposed ice sheet boundary con-
ditions, namely LGM experiments forced by the GLAC-1D (LGM_CAM5_noVeg) or ICE-6G (LGM_CAM5_noVeg_ice6g)
ice-sheet reconstructions (Table 2). On a large scale, using the GLAC-1D ice-sheet reconstruction leads to a smaller LGM JJA
temperature anomaly in the Northern Hemisphere (-6.4°C) than the simulation that includes the ICE-6G ice-sheet reconstruc-
tion (-7.2°C; Figure 5A). Especially in the northeastern Siberian target region the LGM simulation using GLAC-1D ice sheets
is substantially warmer (9.0°C) compared to the simulation using ICE-6G (6.0°C; Table 3). This can only be caused by changes
in the large-scale atmospheric circulation since the simulations are identical apart from the ice sheets over North America and
Eurasia. In line with the PMIP simulations, we find that higher JJA temperatures in the Siberian target region correspond to
specific changes in the 500hPa geopotential height field, with negative anomalies to the southwest and positive anomalies to
the southeast (Figure 5B), and that this stationary wave pattern results in anomalous 500hPa southerly winds into the target
region and a corresponding anomalous northward heat transport almost all the way from 30°N to the North Pole (Figure 5C).
We thus find a high sensitivity of Siberian JJA temperatures with respect to relatively minor changes in the continental ice sheet
geometries, which in turn induce changes in the circumpolar stationary wave pattern and anomalous northward heat transport
in CESM. The similarity of the associated temperature and geopotential height anomaly patterns (wavenumber 2 structure;
Figure 5) with the PMIP-based response (Figures 1 and 3) suggests that this mechanism could also explain part of the spread in
PMIP simulations. The anomalous northward heat transport we see in the stationary waves contributes to reinforce the (clima-
tological) thermal low over Siberia and explains the negative relationship between JJA temperature and JJA surface pressure
anomalies in the Siberian target region, both in the CESM "Continental ice sheets" set of experiments (Table 3) as well as in
the PMIP results (Figure 2C).

The second set of CESM LGM simulations ("Atmospheric model physics"), is comprised of simulations in which different ver-

sions of the atmospheric model were used (CAM4 or CAM5; Table 2). Between the LGM_CAM4_noVeg and LGM_CAM5_noVeg simulations we find changes in the large-scale atmospheric circulation, in particular the stationary waves, and northward heat transport into Siberia (Figure 6) that are broadly similar to the response to different ice sheets as described above. Similar to the analysis of the PMIP models (Figure 3) and the CESM "Continental ice sheets" set of experiments (Figure 5), we find that using different atmospheric model physics can lead to JJA warming (cooling) in the Siberian target region in response to enhanced (decreased) meridional heat transport into northeastern Siberia. Interestingly, if we look in more detail we find that the resulting surface temperature changes in Siberia are more complex in the "Atmospheric model physics" set of experiments than for the experiments described previously. There is warming in some parts of the region, but also cooling in other parts (Figure 6A) and there are differences in the stationary wave pattern and in meridional heat transport. This is possibly related to slight shifts in the centers of action in the geopotential height anomalies and resulting changes in the airmasses that enter the Siberian target region. This highlights the complexity of comparing simulations with different atmospheric model versions that not only differ in their response of the large-scale atmospheric circulation to LGM boundary conditions, but also exhibit different local feedbacks with changes in cloud cover, humidity and pressure, which are directly influenced by, for instance, differences in cloud parameterizations and radiative properties of the atmosphere. This point is further exemplified by the substantial differences between CAM4 and CAM5 in Siberian JJA temperatures and snow cover under PI conditions (Table 3). The models in the PMIP ensemble all differ in the included atmospheric physics and dynamics, thus the described mechansism in this CESM "Atmospheric model physics" set of experiments could as well explain (part of) the spread within the PMIP ensemble.

An important element in the high-latitude climate system is the vegetation-climate feedback. In the PMIP ensemble, 7 out of 17 models include the vegetation-climate feedback (Table 1). However, a systematic difference in simulated JJA LGM temperature anomalies for the Siberian region could not be found when comparing models with vegetation feedback with those that did not include this additional feedback. This doesn't come as a surprise if one considers the relatively small sample size with respect to all the inter-model differences that impact the simulated LGM JJA temperatures. We performed PI and LGM simulations with CESM including and excluding interactive vegetation (the "Interactive vegetation" set; Table 2) to investigate its importance for Siberian temperatures. We find that the vegetation-climate feedback leads to a large LGM JJA cooling over Siberia, which is even more pronounced when using the CAM5 atmospheric model instead of CAM4 (Figure 7 and Table 3). If vegetation is allowed to respond to the changing climate through carbon-nitrogen dynamics, the tree and shrub limits shift south by several degrees of latitude as shown by the leaf area index (Figures 8A and 8C). In CESM, the presence of vegetation, its height as well as its density have a large impact on the surface albedo through the vegetation-albedo feedback: vegetation that protrudes through the snow pack lowers the surface albedo that in turn leads to a positive feedback loop with increasing temperatures, more snow melt, more vegetation growth and an even lower surface albedo. Accordingly, the situation in the CESM simulations including interactive vegetation is such that the cold and snow covered landscape limits vegetation growth and leads to a southward migration of the tree and shrub limits. This relationship between vegetation and snow cover also determines the resulting LGM JJA temperature changes (compare figures 7A and 8C; Table 3). Previous studies also found an important role of vegetation feedbacks in defining LGM Arctic temperatures (Jahn et al., 2005). The impact of interactive

vegetation in CESM is also clearly seen in the PI simulations, resulting in a substantial decrease in the leaf area index with respect to the prescribed values (Figure 8A and 8B) and is in line with the cold bias in modelled Siberian surface temperatures

described by Lawrence et al. (2011) (see also table 3).

Looking at all the experiments in the third set of experiments ("Interactive vegetation", table 2), using different atmospheric model physics (Figure 6) with or without interactive vegetation (Figure 7), we find that the strong cooling in Siberia in the simulation that combines both the different atmospheric model physics and interactive vegetation (LGM_CAM5_Veg; Figure 7B), is not readily explained as a linear combination of the two individual effects. This is true for Siberian JJA temperatures,

but also for other key climate variables (Table 3). It should be noted that the simulations with the lowest JJA LGM temperatures in table 3 are in fact the ones with the highest precipitation rates (not only in JJA, but also in the annual mean; not shown). This all shows the complexity of the response to a combination of factors, in this case changes in large-scale atmospheric circulation, local atmospheric processes and local land-surface processes. It is to be expected that the response of individual PMIP simulations is similarly complex.

**4   Concluding remarks**

From a climate model perspective, LGM JJA temperatures in northeastern Siberia appear highly susceptible to changes in the imposed boundary conditions, included feedbacks and processes, and to the model physics of the different climate model components; much more so for Siberia than for any other region. This becomes apparent from the comparison of 17 different PMIP2 and PMIP3 LGM experiments, as well as from three sets of CESM sensitivity experiments. The spread in Siberian JJA

LGM temperature anomalies in the CESM ensemble is ~20°C, which is comparable to the inter-model spread of ~24°C found in the PMIP simulations. The main cause appears to be that relatively small changes in the continental ice sheets or model physics can lead to large changes in meridional atmospheric heat transport related to changes in the circumpolar atmospheric stationary wave pattern, in line with Ullman et al. (2014) and Liakka and Lofverstrom (2018). Local snow-albedo and vegetation-climate feedbacks strongly amplify the Siberian JJA temperature change. Recently, Schenk et al. (2018) showed

that the spatial resolution of the atmospheric model is key to obtaining realistic glacial temperature anomalies. However, we do not find any correlation between atmospheric model resolution and Siberian JJA LGM temperature anomalies (Table 1), despite having some models with a resolution very similar to the one used by Schenk et al. (2018). We note, however, that we did not perform a dedicated sensitivity experiment changing only the spatial resolution while keeping all other factors the same.

In most of the examined PMIP LGM simulations Siberia receives less precipitation; however, we don't find indications that the buildup of a Siberian ice sheet was hampered by the absence of precipitation. On the contrary, in both the PMIP ensemble as our CESM experiments we find that local precipitation and JJA temperature changes are not significantly correlated, while cooler summers are strongly correlated to a higher snow cover, suggesting that a cold climate would be associated with a perennial snow cover. Neither do we find support for the notion that changes in large-scale atmospheric stationary wave patterns drive Siberian JJA temperatures directly through local cloud changes.

Although situated at high northern latitudes, geological evidence suggests that Siberia was covered by continental ice sheets during some glacial periods, but remained largely ice free during others, for instance the last glacial period including the LGM.

5  Increased atmospheric dust deposition and a precipitation-shadow cast by the Eurasian Ice Sheets to the west are often listed as possible causes; however, such mechanisms cannot readily explain the absence of a Siberian ice sheet in some glacial periods, but its presence in others, or conform with the independent reconstructions of Siberian LGM summer temperatures close to present-day values (Meyer et al., 2017). This is suggesting that these processes are likely only part of the story, and here we argue for the importance of changes in meridional atmospheric heat transport and the configuration of the northern hemisphere

10  continental ice sheets in order to understand the geological evidence. The combination of these factors, accompanied by local feedbacks can lead to strongly divergent summer temperatures in the region, which during some glacial periods could have been sufficiently low to allow for the buildup of an ice sheet, while during other glacials, above-freezing summer temperatures might have prevented a multi-year snow-pack, and hence an ice sheet, from forming. Finally, this high sensitivity of Siberian LGM summer temperatures in different climate models will present a major challenge in future modelling efforts using coupled ice-sheet-climate models.

*Data availability.* For the PMIP experiment results see https://pmip2.lsce.ipsl.frandhttps://pmip3.lsce.ipsl.fr for further details and references. Results from the CESM sensitivity experiments can be obtained from the authors.

*Author contributions.* P. B. and I. R. designed the study. P. B. performed the CESM sensitivity experiments and analysed the PMIP and CESM experiments. P. B. wrote the manuscript. I. R. reviewed the literature for geological and climatological reconstructions. All authors participated in the discussion of the results and the manuscript, and provided feedback and comments.

*Competing interests.* The authors declare no conflict of interest.

*Acknowledgements.* This study is a contribution to the PalMod project funded by the German Federal Ministry of Education and Science (BMBF). The climate model simulations were carried out on the supercomputer of the Norddeutscher Verbund für Hoch- und Höchstleistungsrechnen (HLRN). We thank Pedro diNezio for providing us with CESM LGM initial and boundary conditions.

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

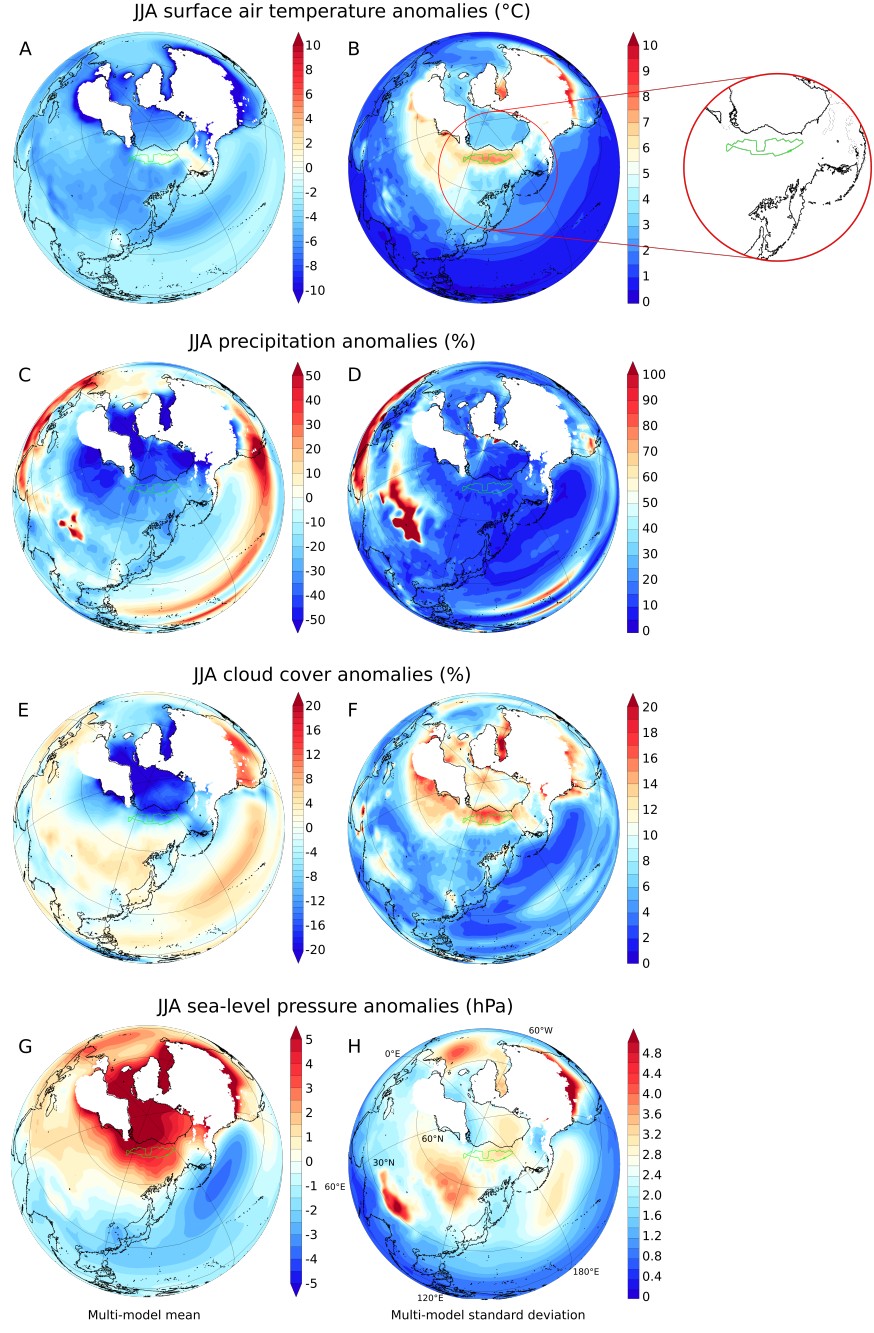

**Figure 1.** The PMIP2 and PMIP3 multi-model mean (left panels) and multi-model standard deviation (right panels) in LGM JJA climate anomalies. A-B: temperature anomalies (°C). C-D: precipitation anomalies (%). E-F: cloud cover anomalies (%); G-H: sea-level pressure anomalies (hPa). All anomalies are calculated with respect to PI. Note that regions covered by continental ice sheets during the LGM have been masked out. The green contour (shown in magnification in the top-right) shows the Siberian target region defined here as the region in which the PMIP multi-model standard deviation is larger than 7°C. The LGM coastlines are given in black.

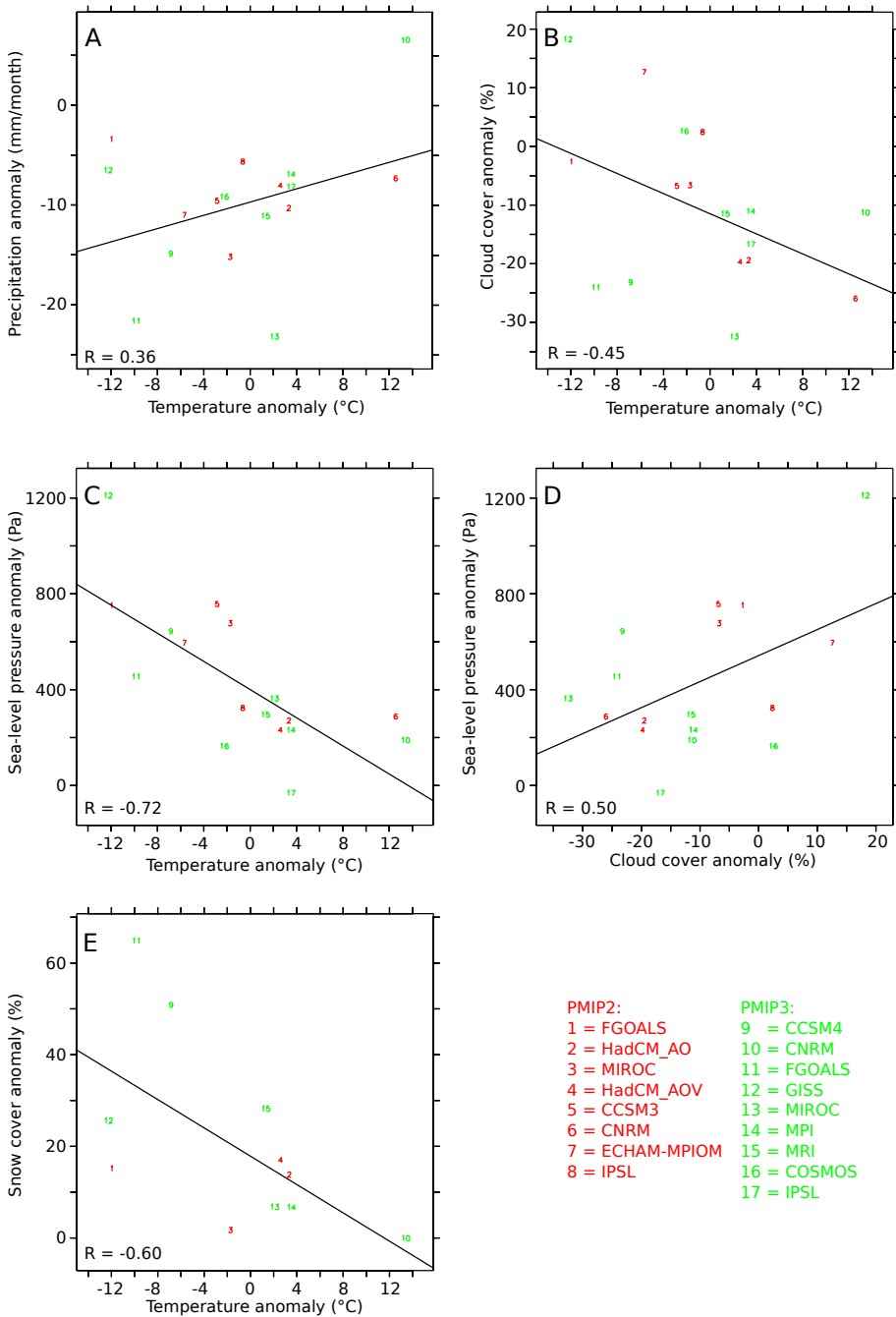

**Figure 2.** PMIP2 and PMIP3 LGM JJA climate anomalies averaged over the northeast Siberian target region. Red (green) numbers refer to the individual PMIP2 (PMIP3) experiments listed in the lower right. A: Precipitation anomalies ($\mathrm{mm\ month}^{-1}$) versus temperature anomalies ($^{\circ}$C). B: Cloud cover anomalies (%) versus temperature anomalies ($^{\circ}$C). C: Sea-level pressure anomalies (Pa) versus temperature anomalies ($^{\circ}$C). D: Sea-level pressure anomalies (Pa) versus cloud cover anomalies (%). E: Snow cover anomalies (%) versus temperature anomalies (K). Black lines show linear fit and the R-value (Pearson correlation coefficient) as listed in the lower left corners of the different subfigures. R-values above 0.49 or below -0.49 indicate a significant correlation (p<0.05; t-test).

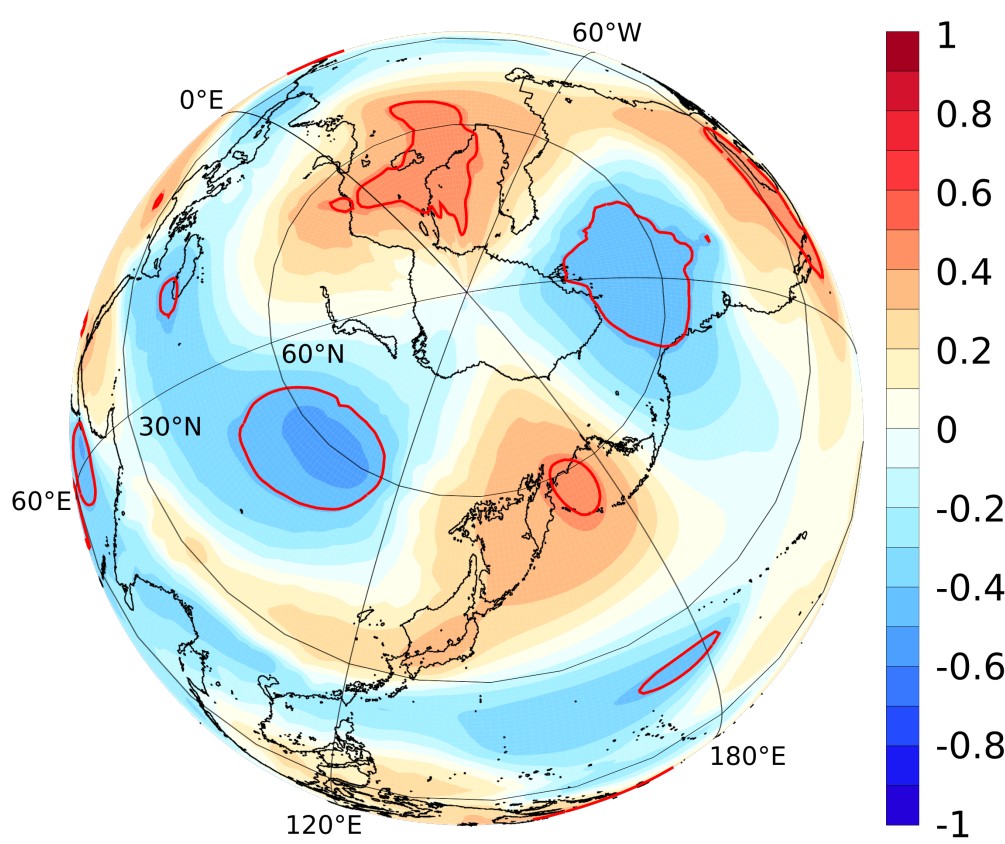

**Figure 3.** PMIP2 and PMIP3 linear correlations between JJA 500 hPa stationary wave geopotential height anomalies at any given location and JJA temperature anomalies averaged over the Siberian target region (see Figure 1 for the definition). Anomalies are calculated with respect to PI, and zonal mean geopotential height fields are subtracted before calculating the anomalies. The red contours bound the areas for which the correlation is significant (p< 0.1). The LGM coastlines are given in black.

## JJA surface air temperature anomalies (°C)

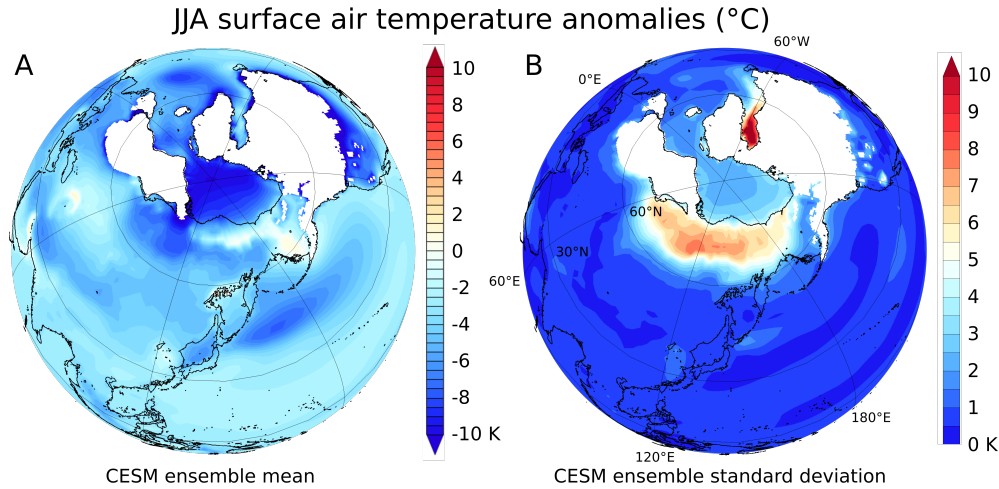

**Figure 4.** CESM ensemble mean (A) and ensemble standard deviation (B) of LGM JJA temperature anomalies ($^{\circ}$C). Note that regions covered by continental ice sheet during the LGM have been masked out. The LGM coastlines are given in black.

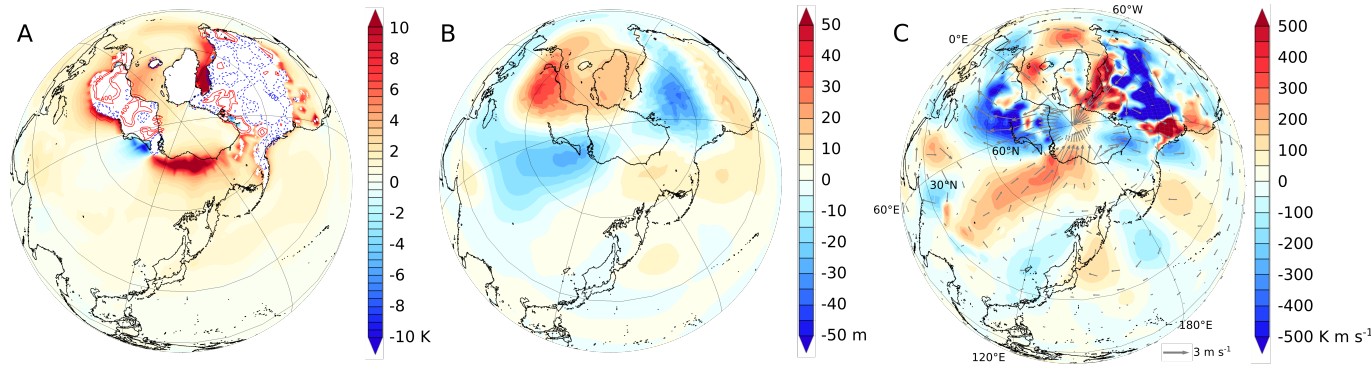

**Figure 5.** Impact of the prescribed LGM ice-sheet topography (GLAC-1D versus ICE-6G) on simulated LGM climate anomalies during the boreal summer season (JJA). Results are shown as the CESM experiment LGM_CAM5_noVeg minus LGM_CAM5_noVeg_ice6g. A: near-surface temperature anomalies (K). B: 500hPa geopotential height anomalies (m; anomalies calculated after subtracting the zonal mean). C: vertically averaged meridional sensible heat transport anomalies ($\mathrm{Kms^{-1}}$; shading). Vectors in panel C show 500 hPa wind anomalies ($\mathrm{ms^{-1}}$). In panel A regions covered by continental ice sheets during the LGM have been masked out. The red (blue) contours in panel A depict positive (negative) differences in ice sheet height (m) between the GLAC-1D and ICE-6G reconstructions ( GLAC-1D - ICE-6G; 300 m contour interval). The LGM coastlines are given in black.

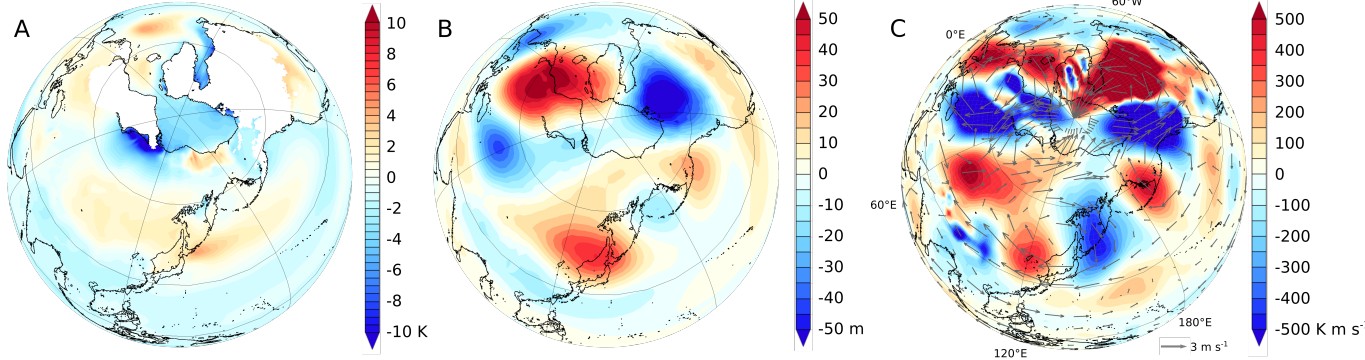

**Figure 6.** Impact of using different atmospheric models (CAM5 versus CAM4) on simulated LGM climate anomalies during the boreal summer season (JJA). Results are shown as LGM-PI anomalies for LGM_CAM5_noVeg minus LGM_CAM4_noVeg. A: near-surface temperature anomalies (K). B: 500 hPa stationary wave geopotential height anomalies (m; anomalies calculated after subtracting the zonal mean). C: vertically averaged meridional sensible heat transport anomalies ($Kms^{-1}$; shading). Vectors in panel C show 500 hPa wind anomalies ($ms^{-1}$). In panel A regions covered by continental ice sheets during the LGM have been masked out. The LGM coastlines are given in black.

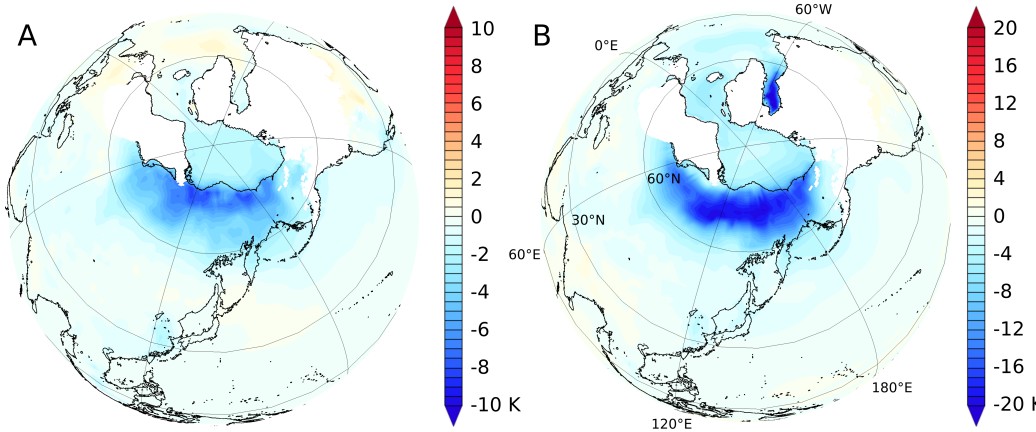

**Figure 7.** JJA LGM temperature anomalies showing the impact of introducing vegetation-climate feedbacks. Results are shown as LGM-PI anomalies for CAM4 (A; LGM_CAM4_Veg – LGM_CAM4_noVeg) and CAM5 (B; LGM_CAM5_Veg – LGM_CAM5_noVeg). Regions covered by continental ice sheets during the LGM have been masked out. The LGM coastlines are given in black. Note the different scaling used for the two panels.

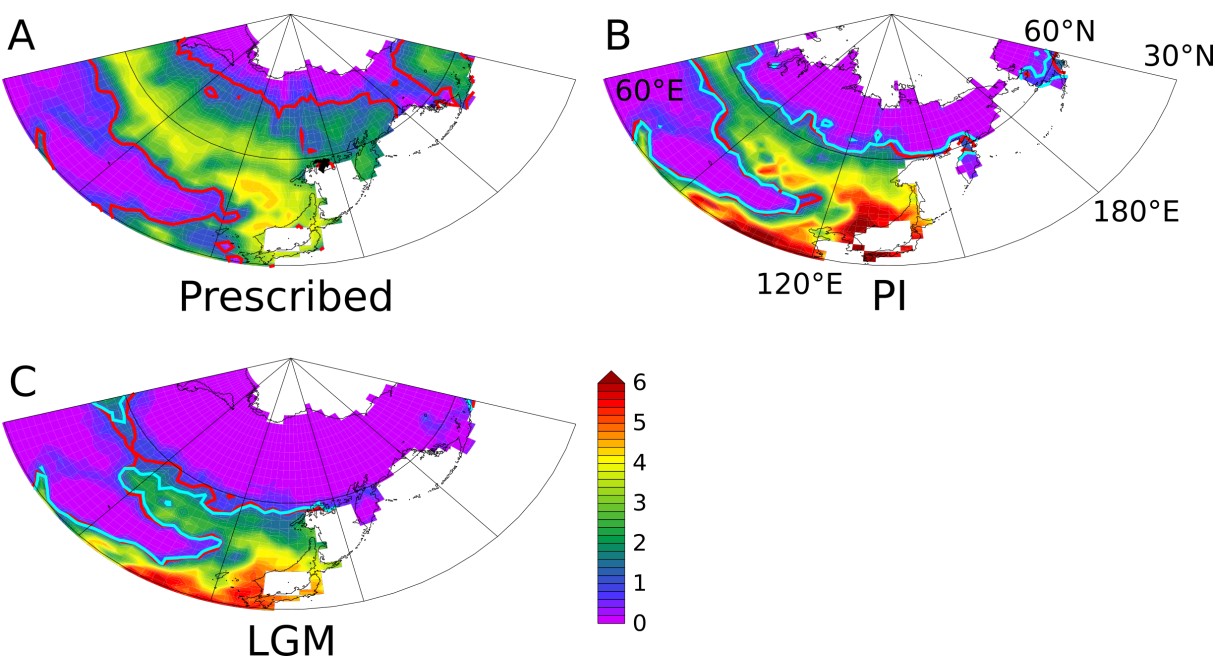

**Figure 8.** Leaf area index $(\mathrm{m}^2\mathrm{m}^{-2})$ in northeastern Asia as prescribed in the simulations without interactive vegetation (A), and as simulated in the pre-industrial (B) and LGM (C) CAM4_Veg experiments including interactive vegetation. Contours give the leaf area index of 1 $\mathrm{m}^2\mathrm{m}^{-2}$ (red for CAM4 and light blue for CAM5).

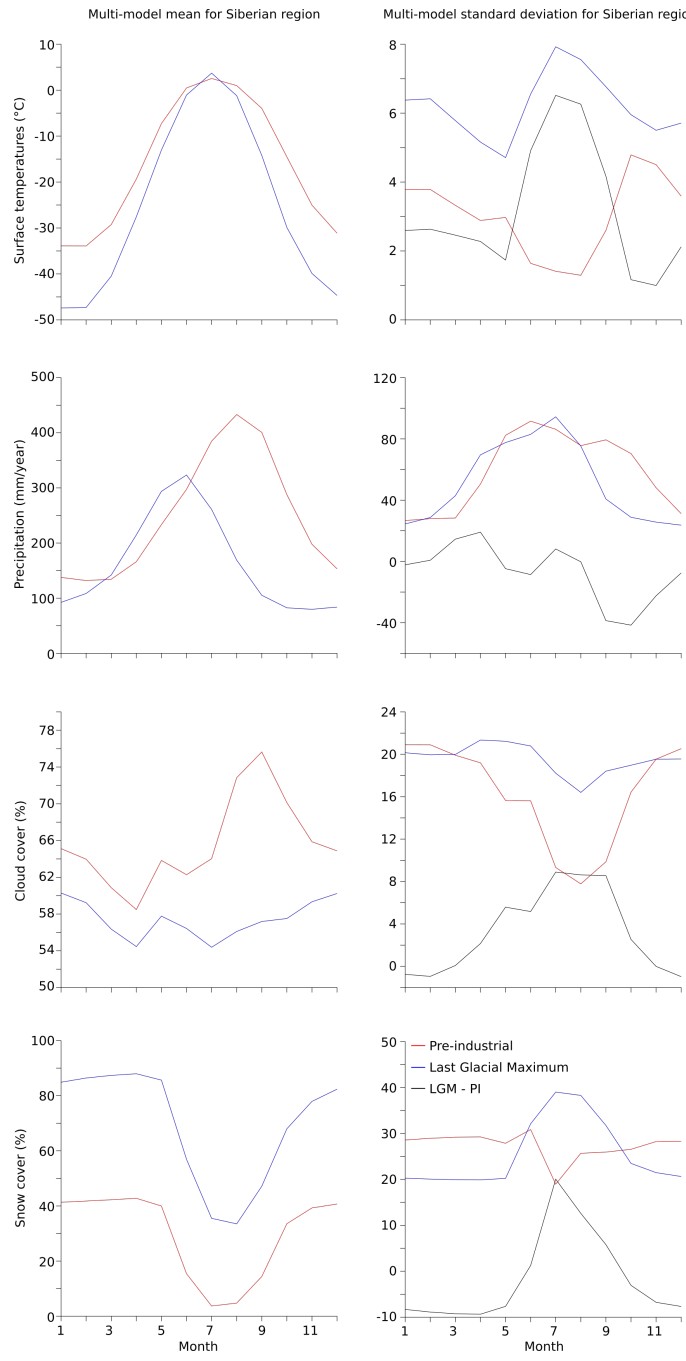

**Figure A1.** PMIP2 and PMIP3 multi-model mean (left panels) and multi-model standard deviation (right panels) seasonal cycles of selected variables for PI (red), LGM (blue) and LGM anomalies (LGM - PI; black). Mean and standard deviation calculated for the Siberian target region. Top row: temperatures ($^{\circ}$C); Second row: precipitation ($\mathrm{mm\,yr^{-1}}$); Third row: cloud cover (%); Bottom row: snow cover (%).

**Table 1.** List with PMIP2 and PMIP3 climate models included in the analysis with details on grid resolution and usage of interactive vegetation. In the last column the simulated LGM JJA surface temperature anomaly (K) in the Siberian target region with respect to the pre-industrial is given for reference. The following abbreviations are used: Atm (atmospheric grid resolution), Ocn (ocean grid resolution), L (number of levels in the vertical). See https://pmip2.lsce.ipsl.fr and https://pmip3.lsce.ipsl.fr for further details and references.

| Model | Institution | Grid Resolution | Interactive vegetation | PMIP phase | $\Delta T$ |
|---|---|---|---|---|---|
| CCSM3 | National Center for Atmospheric Research, USA | Atm: 128 x 64 x L26<br>Ocn: 320 x 384 x L40 | No | 2 | -3.0 |
| CNRM-CM3.3 | Centre National de Recherches Meteorologiques, France | Atm: 256 x 128 x L31<br>Ocn: 362 x 292 x L42 | No | 2 | 12.4 |
| ECHAM5-MPIOM | Max Planck Institute for Meteorology, Germany | Atm: 96 x 48 x L19<br>Ocn: 120 x 101 x L40 | Yes | 2 | -5.8 |
| FGOALS1.0_g | LASG/Institute of Atmospheric Physics, China | Atm: 128 x 60 x L26<br>Ocn: 360 x 180 x L33 | No | 2 | -12.1 |
| HadCM3_AO | UK Met Office Hadley Centre, UK | Atm: 96 x 72 x L19<br>Ocn: 288 x 144 x L20 | No | 2 | 3.2 |
| HadCM3_AOV | UK Met Office Hadley Centre, UK | Atm: 96 x 72 x L19<br>Ocn: 288 x 144 x L20 | Yes | 2 | 2.4 |
| IPSL-CM4_v1 | Institut Pierre Simon Laplace, France | Atm: 96 x 72 x L19<br>Ocn: 182 x 149 x L31 | No | 2 | -0.8 |
| MIROC3.2.2 | Center for Climate System Research, JAMSTEC, Japan | Atm: 128 x 64 x L20<br>Ocn: 256 x 192 x L43 | No | 2 | -1.9 |
| CCSM4 | National Center for Atmospheric Research, USA | Atm: 288 x 192 x L26<br>Ocn: 320 x 384 x L60 | Yes | 3 | -7.0 |
| CNRM-CM5 | CNRM - C. Européen de Rech. Formation Avancée Calcul Sci. | Atm: 256 x 128 x L31<br>Ocn: 362 x 292 x L42 | No | 3 | 13.2 |
| COSMOS-ASO | Max Planck Institute for Meteorology, Germany | Atm: 96 x 48 x L19<br>Ocn: 120 x 101 x L40 | Yes | 3 | -2.4 |
| FGOALS_g2 | ASG/Institute of Atmospheric Physics, China | Atm: 128 x 60 x L26<br>Ocn: 360 x 180 x L30 | No | 3 | -10.0 |
| GISS-E2-R | NASA Goddard Institute for Space Studies | Atm: 144 x 90 x L40<br>Ocn: 288 x 180 x L32 | No | 3 | -12.4 |
| IPSL-CM5A-LR | Institut Pierre Simon Laplace, France | Atm: 96 x 96 x L39<br>Ocn: 182 x 149 x L31 | Yes | 3 | 3.4 |
| MIROC-ESM | Center for Climate System Research, JAMSTEC, Japan | Atm: 128 x 64 x L80<br>Ocn: 256 x 192 x L44 | Yes | 3 | 1.9 |
| MPI-ESM-P | Max Planck Institute for Meteorology, Germany | Atm: 196 x 98 x L47<br>Ocn: 256 x 220 x L40 | Yes | 3 | 3.4 |
| MRI-CGCM3 | Meteorological Research Institute (MRI) | Atm: 320 x 160 x L48<br>Ocn: 364 x 368 x L51 | No | 3 | 1.1 |

**Table 2.** List of simulations included in the three sets of CESM LGM experiments and the PI reference simulations. The following abbreviations are used: noVeg = No interactive vegetation; Veg = Including interactive vegetation; PI = pre-industrial; LGM = Last Glacial Maximum; CAM4/5 = Community Atmosphere Model version 4 or version 5; GLAC-1D = GLAC-1D ice sheet reconstruction (Ivanovic et al., 2016); ice6g = ICE-6G ice sheet reconstruction (Peltier et al., 2015).

| Experiment set | Experiment name | Atmospheric model | Interactive vegetation | Boundary conditions | LGM ice-sheet reconstruction |
|---|---|---|---|---|---|
| PI reference simulations | PI_CAM4_noVeg | CAM4 | No | PI | |
| | PI_CAM5_noVeg | CAM5 | No | PI | |
| | PI_CAM4_Veg | CAM4 | Yes | PI | |
| | PI_CAM5_Veg | CAM5 | Yes | PI | |
| Continental ice sheets | LGM_CAM5_noVeg | CAM5 | No | LGM | GLAC-1D |
| | LGM_CAM5_noVeg_ice6g | CAM5 | No | LGM | ICE-6G |
| Atmospheric model physics | LGM_CAM4_noVeg | CAM4 | No | LGM | GLAC-1D |
| | LGM_CAM5_noVeg | CAM5 | No | LGM | GLAC-1D |
| Interactive vegetation | LGM_CAM4_noVeg | CAM4 | No | LGM | GLAC-1D |
| | LGM_CAM4_Veg | CAM4 | Yes | LGM | GLAC-1D |
| | LGM_CAM5_noVeg | CAM5 | No | LGM | GLAC-1D |
| | LGM_CAM5_Veg | CAM5 | Yes | LGM | GLAC-1D |

**Table 3.** Simulated CESM PI and LGM climatic conditions in the Siberian target region. For the abbreviations see table 2. Note that LGM JJA sea level pressure show here have been corrected for LGM to PI differences in global mean sea level pressure.

| | Experiment name | JJA temperatures °C | JJA precipitation mm month$^{-1}$ | JJA cloud cover % | JJA sea level pressure hPa | Minimum snow cover % | JJA snow cover % |
|---|---|---|---|---|---|---|---|
| PI reference simulations | PI_CAM4_noVeg | 8.0 | 5.5 | 55.1 | 1009 | 1.3 | 10.9 |
| | PI_CAM5_noVeg | 10.7 | 3.1 | 65.7 | 1012 | 0.0 | 1.7 |
| | PI_CAM4_Veg | 6.5 | 6.7 | 54.2 | 1009 | 2.8 | 23.1 |
| | PI_CAM5_Veg | 8.4 | 6.5 | 62.4 | 1013 | 0.4 | 13.9 |
| LGM simulations | LGM_CAM4_noVeg | 8.5 | 4.2 | 45.4 | 1010 | 0.6 | 5.8 |
| | LGM_CAM5_noVeg | 9.0 | 7.4 | 62.6 | 1011 | 0.6 | 4.5 |
| | LGM_CAM4_Veg | 1.4 | 10.5 | 46.8 | 1011 | 16.8 | 43.7 |
| | LGM_CAM5_Veg | -12.1 | 20.3 | 70.4 | 1019 | 100.0 | 100.0 |
| | LGM_CAM5_noVeg_ice6g | 6.0 | 11.4 | 58.2 | 1013 | 2.5 | 9.6 |