# Peer review of "Hypersensitivity of glacial summer temperatures in Siberia"

_Climate of the Past, 2019_

## Referee Comment (RC1) · Anonymous Referee #1 · 31 Jul 2019

Bakker et al. look to understand the mechanisms responsible for Siberian climate at the LGM. To do so, they use a combination of PMIP2/3 simulations and CESM1 sensitivity tests. The authors find that the Siberian region has a large temperature and precipitation spread among models. Using their CESM1 sensitivity tests, Bakker et al. explore the sensitivity of the Siberian region to model physics, ice sheet configuration, and vegetation response. They find that the Siberian temperature response is most significantly influenced by the vegetation, especially when using CAM5, but ice sheet geometry and model physics are also important. Overall, this is a nice study that I believe will be a valuable contribution to understanding climate in a largely overlooked region at the LGM. However, I have a few questions about the model configurations and would like a bit more detailed exploration of mechanisms before publication.

[Figure]

Major Comments:

Additional information about the model setup is required. How was the original LGM simulation, from which these experiments were branched, configured? This is important, because as the authors find, the climate produced by CAM4 and CAM5 can be quite different. Therefore, despite branching from a previous run, I am not convinced that 200 years of spin up is sufficient. Including top-of-atmosphere energy imbalance would provide a first order estimate of how close these simulations are to equilibrium. Also, are 30 year averages enough to produce true climatologies in this region? There are a lot of decadal oscillations that can impact climate for long periods (e.g. Deser et al. 2012). I don't think that this will significantly change results, but I do recommend a quick comparison with a longer average, such as 50 years, to make sure. Finally, how were the CLM4 cases with "interactive vegetation" spun up? If not spun-up properly, it can take hundreds of years for the carbon cycle to come into equilibrium, which could impact your vegetation distribution.

Limiting the analyses to JJA limits the mechanistic understanding. Are you sure that the summer changes are mainly a result of summer processes? Also, a more rigorous exploration of the local radiative effects versus heat transport would be useful. For example, albedo and cloud radiative forcings would be more insightful than snow and cloud cover.

The authors argue for the necessity of additional CESM simulations based in part on the number of variables available for analysis from the PMIP simulations, but proceed to explore only basic outputs from their CESM experiments. Additional analyses to explore why the temperature changes in CESM with different configurations is warranted. At a minimum, areas of perennial snow cover are worth including. What about sea ice? Maybe a PDD and/or energy balance calculation would be insightful. With additional information, the authors could make a much more significant statement about which simulations would produce an ice sheet in Siberia at the LGM. From there, additional model assessment with proxies is possible. Are the models that produce a Siberian ice

sheet too cold (probably) or too wet, etc. . .? What does this suggest about Siberian climate at the LGM?

Specific:

P1 Line 20: Further south than 50°N in many locations in North America. P1 Line 21: Much of Alaska also did not have ice. P1 Line 25: Didn't some of these modeling studies limit their ice domain to exclude Siberia? Double check. P2 Line 2: Citation for the sea level statement? P2 Line 30: This dust feedback is mentioned in earlier (e.g. Mahowald et al., 1999; Ganopolski et al., 2010). What about the direct radiative effect of dust (e.g. Schneider et al., 2006)? P4 Line 1: Link is messed up. P4 Line 10: Should be 1.9x2.5° P5 Line 16: Shouldn't this citation be for an ice sheet reconstruction paper? Peltier et al. (2015) maybe? P5 Line 34: Not sure that ensemble is the correct word. P6 Line 5: Need to spell out LGM_CAM5_noVeg first. P6 Line 14: Why not look at the snow cover in the model? P6 Line 4: Cloud radiative forcing would be more insightful. P6 Line 13: Did you analyze CCSM3, as used in Liakka et al. (2016), to better understand this discrepancy? Could use a bit of additional discussion. Figure 1: Make the continental outlines thicker. Figure 2: Darker green would make it easier to see. Figure 3: Add winds and/or height anomalies to better highlight the circulation changes. P10 Line 16: Why not plot the same variables as in the PMIP runs with CESM? P10 Line 22: How is surface roughness over the ice sheets configured? The results of Brady et al. (2013) suggest that this is important. P10 Line 8-2?: It would be great to plot some of the differences mentioned. . . P11 Line 20: How do you define vegetation density? P11 Line 22: This vegetation feedback has been found to be important for Arctic climate before (e.g. Jahn et al., 2005; Tabor et al., 2014). Figure 5 A: There must be a strong local feedback in Siberia. Maybe plot snow cover or albedo? P11 Line 23: Does this mean the vegetation dies? P11 Line 26: Does Lawrence et al. (2011) discuss this Arctic LAI issue? Figure 6: How were your PI runs configured for your LGM-PI anomalies? The vectors are very hard to see in panel C. Please change the color. Figure 7: Extend the temperature range in panel B.

---

## Referee Comment (RC2) · Anonymous Referee #2 · 31 Jul 2019

Geological evidence has shown that Siberia was partially glaciated during some glacial states while it kept mostly ice-free during others. Different previous studies have explored several potential explanations for these differences but a consensus is still lacking. Bakker et al. show that the ensemble of climate model experiments from PMIP2 and PMIP3 shows a very large spread in their simulated glacial summer (JJA) temperatures for the last glacial maximum (LGM) over Siberia. Bakker et al. argue that the large model spread could be an indication for a real "hypersensitivity" of glacial summer temperatures over Siberia, and hence regional glaciation itself. To explore some of the possible factors which may result in climatic differences over Siberia, they conduct several sensitivity simulations with CESM and show that the spread in simulations re-

sulting from different ice sheet heights, vegetation feedback or changes in atmospheric physics of CAM4/5 can cause an equally large spread (∼20 K) as the PMIP model ensemble (∼24 K).

Overall, the manuscript is very well written and provides interesting insights into the problem of glacial summer temperature hypersensitivity and how it might explain the absence or presence of glaciation in Siberia during different glacials. However, the potential reasons for what may cause the large simulated temperature spread over Siberia could be explored in a bit more detail. I generally recommend publication in Climate of the Past after adding some more analysis to explain the summer temperature discrepancies.

General comments:

The study is very well written and presents very interesting and important aspects to better understand the possibly real "hypersensitivity" of the Siberian climate during glacials as well as the behaviour of models. Regarding the analysed variables in the manuscript, it is a bit difficult to understand whether local radiative processes (e.g. what about albedo, spring snow cover and lagged warming?) or large-scale temperature advection play a major role for the temperature spread – or both. Because Siberia builds up a spatially widespread thermal low during summer, the correlation between summer temperature and SLP can be expected to be mainly temperature driven. Increasing temperature will hence cause lower SLP which then can increase horizontal advection into Siberia. Consequently, changes in SLP would be rather a feedback to the warming (or cooling) and not the mechanism which causes the effect.

I also wonder whether the correlation in Fig. 3 is really statistically significant in terms of field significance given the low spatial degrees of freedom of SLP and that the relatively small regions with statistically significant correlations might be just those which are allowed to be significant by chance. In general, I would rather expect that the large-scale gradients in the pressure and temperature field e.g. relative to the Arctic and

Tropics are important for temperature advection into Siberia. It would be interesting to see some analysis of the large-scale wind fields or pressure gradients and how different they are with respect to the model spread e.g. of the warmest vs. coldest PMIP member. I was also wondering if large-scale teleconnections might be very different for very warm vs. very cold simulations of Siberian summer temperatures (e.g. a one-point-correlation map of the averaged Siberian SLP and temperature with the northern hemisphere SLP and temperature).

Regarding the large temperature spread over Eurasia, I was also wondering whether there is a potential link between warm and cold model experiments and the used atmospheric resolution (see below). In any case, the paper would strongly gain from a bit more detailed analysis and discussion of these aspects while the rest of the paper is very well written and does not require notable changes with exception of clarifying the sections about the role of the thermal low.

Specific comments:

Title of the paper: Maybe be more specific and write "glacial summer temperatures"?

Page 2, line 2: Due to the quite shallow Arctic shelf, sea-level changes during the glacial lead to quite large changes in additionally exposed land during low level stands along the Arctic and Siberian coast. During summer, the additional landmass clearly increases the area which can heat up strongly during boreal summers with 24 hours of daylight. I could imagine that such an effect would be higher in models with a high horizontal resolution. It would be very interesting if you could add some information in the manuscript about individual ensemble members if there are indications that their differences in atmospheric resolution lead to systematic differences in Siberian temperatures.

In this context, there is one recent example where a very coarse resolution simulation has been repeated with the same ocean state and external forcing but using a 4x higher atmospheric resolution with CESM1 (Schenk et al. 2018) for the late glacial. In

their supplementary figure 4, they show that a much higher atmospheric resolution with CESM1 predicts considerably warmer summers during the Younger Dryas stadial over Eurasia and Siberia compared to the coarse resolution simulation with CCSM3 despite using the same ocean state. They argue that atmospheric blocking in response to the Fennoscandian Ice Sheet (among other reasons) leads to warmer Eurasian summers. They show that the blocking and hence warmer summers are only captured at high resolution. Is this also the case for the warmest vs. coldest PMIP members?

Given the very strong difference in simulated summer temperatures at a different model resolution by Schenk et al. (2018) and the very important results of other studies concerning the atmospheric flow disturbance by ice sheets (as already cited by the authors on page 3), I would suggest to add a paragraph about whether atmospheric resolution differences in the presence of large continental ice sheets can partly explain the spread of warming or cooling over Siberia.

Regarding the exposed Arctic shelf during stadials: Is there any geological evidence that glaciations in Siberia might correlate with periods of higher sea-level stands (less exposed Arctic shelf and possibly cooler summers with a weaker thermal low and less advection)?

Page 2, line 20: Can you give an example which one is good and possibly why?

Page 5, line 1: Components of GLAC-1D have been published in different papers. Please add here the reference of the complete version which is Ivanovic et al. (2016).

Page 5, line 3: Figure 5A is too small to see the important differences in ice sheet heights.

Page 5, line 27: The green contour line is not visible. Please add in addition the coordinates for the target region in the manuscript (for the analysed 1°x1° grid).

Page 6, lines 13-14: Regarding "…could be a consequence of local temperature changes…": This is quite certain as the low pressure over Siberia during summer

is a thermal low and not a dynamic low. The sentence should be modified accordingly.

Page 6, lines 15-16: The link to the Asian monsoon region and possibly other large-scale teleconnections are very important and should be explored a bit more in the manuscript.

Page 6, lines 17-25: The paragraph should be clarified with respect to the low being a thermal low. It appears odd to argue here that a deepening of the low-pressure cell over central Asia (it is not really a cell but rather a diffuse area) should control the amount of warming in Siberia when the deepening of the low is driven by the warming. This might be rather a positive feedback where warming increases convection which lowers the pressure which increases horizontal advection. This implies that another process causes the warming and the change in SLP is only a feedback. Please re-write accordingly.

Page 10, lines 21-22: It would be interesting to get a number for the overall temperature change of the northern hemisphere in response to using a different ice sheet in CESM.

Page 10, lines 24-25: This again is due to the thermal low which has to deepen with increasing temperature due to an increase in the rise of warm air.

Page 10, line 31: The similarity of the spatial anomaly pattern for temperature and SLP can be expected for the behaviour of the thermal low in summer. There has to be another reason for the warming first and the SLP change cannot be the mechanism but rather a positive feedback.

Page 15, line 17: Please add a concluding paragraph about which model configuration for CESM (and e.g. which ice sheet) would be plausible for the LGM (no glaciation in Siberia) and why. In this context, can you give some examples about which PMIP models would be plausible for the LGM and absence of Siberian glaciation and which not and why?

Figure 1: The green contour in panel B is not visible.

Figure 2: Please strongly increase the size of numbers in the figure as well as the axis description.

Figure 3: Are the significant areas really statistically significant globally or only by chance? Given that the correlations may rather represent the thermal low, I'm not sure how this figure helps to understand the spatial spread over Siberia. Pressure gradients and teleconnections might be more suitable as they would represent how the changes of the thermal low interact with remote regions.

Figure 8: The red and blue for CAM4/5 is very difficult to see.

Table 1: It would be important to add a column here with the temperature difference LGM minus PI over Siberia for each model simulation to identify which models are unusually warm/cold. This would make it easy for others to further explore why which models differ from others. In this way, a potential dependency on the model resolution could be easily identified.

Table 2: Also here the temperature difference LGM minus PI over Siberia would be interesting.

Additional references:

Ivanovic, R. F. et al. Transient climate simulations of the deglaciation 21—9 thousand years before present (version 1)—PMIP4 core experiment design and boundary conditions. Geosci. Model Dev. 9, 2563–2587 (2016).

Schenk, F. et al. Warm summers during the Younger Dryas cold reversal. Nature Commun. 9:1634 (2018).

———————————————

---

## Author Comment (AC2) · 13 Sep 2019

Geological evidence has shown that Siberia was partially glaciated during some glacial states while it kept mostly ice-free during others. Different previous studies have explored several potential explanations for these differences but a consensus is still lacking. Bakker et al. show that the ensemble of climate model experiments from PMIP2 and PMIP3 shows a very large spread in their simulated glacial summer (JJA) temperatures for the last glacial maximum (LGM) over Siberia. Bakker et al. argue that the large model spread could be an indication for a real "hypersensitivity" of glacial summer temperatures over Siberia, and hence regional glaciation itself. To explore some of the possible factors which may result in climatic differences over Siberia, they conduct several sensitivity simulations with CESM and show that the spread in simulations resulting from different ice sheet heights, vegetation feedback or changes in atmospheric physics of CAM4/5 can cause an equally large spread ($\sim$20 K) as the PMIP model ensemble ($\sim$24 K).

Overall, the manuscript is very well written and provides interesting insights into the problem of glacial summer temperature hypersensitivity and how it might explain the absence or presence of glaciation in Siberia during different glacials. However, the potential reasons for what may cause the large simulated temperature spread over Siberia could be explored in a bit more detail. I generally recommend publication in Climate of the Past after adding some more analysis to explain the summer temperature discrepancies.
**We thank the reviewer for the kind words and for having a critical look at the manuscript.**

General comments:
The study is very well written and presents very interesting and important aspects to better understand the possibly real "hypersensitivity" of the Siberian climate during glacials as well as the behaviour of models. Regarding the analysed variables in the manuscript, it is a bit difficult to understand whether local radiative processes (e.g. what about albedo, spring snow cover and lagged warming?) or large-scale temperature advection play a major role for the temperature spread – or both. Because Siberia builds up a spatially widespread thermal low during summer, the correlation between summer temperature and SLP can be expected to be mainly temperature driven. Increasing temperature will hence cause lower SLP which then can increase horizontal advection into Siberia. Consequently, changes in SLP would be rather a feedback to the warming (or cooling) and not the mechanism which causes the effect.
**We agree with the reviewer that it is difficult to disentangle local versus large-scale effects on Siberian temperatures. This is especially true when doing so through the analysis of sea-level pressure fields. Fortunately we have found out that for all but one PMIP2/3 LGM simulation also geopotential height fields are available, and this makes for a more direct line of arguments and, in our opinion, a more convincing analysis to showe that changes in the large-scale, circumarctic atmospheric circulation are indeed the cause of the spread in simulated Siberian JJA temperatures.**
**In the updated manuscript we will show PMIP results for geopotential height anomalies at 500hPa (with the zonal mean removed), and together with the existing CESM geopotential height results we argue that both clearly show large-scale anomaly patterns that resemble a classical stationary wave pattern and therefor indicate changes in the large-scale atmospheric circulation. Based on CESM results we already showed that as a result of these circulation changes, meridional heat transport into the region under discussion increases. These circumarctic patterns are unlikely to be caused by local Siberian temperature changes, but rather are the cause of the Siberian temperature changes to which in turn local sea-level pressure changes provide a feedback.**

I also wonder whether the correlation in Fig. 3 is really statistically significant in terms of field significance given the low spatial degrees of freedom of SLP and that the relatively small regions with statistically significant correlations might be just those which are allowed to be significant by chance. In general, I would rather expect that the large-scale gradients in the pressure and temperature field e.g. relative to the Arctic and Tropics are important for temperature advection into Siberia. It would be interesting to see some analysis of the large-scale wind fields or pressure gradients and how different they are with respect to the model spread e.g. of the warmest vs. coldest PMIP member. I was also wondering if large-scale teleconnections might be very different for very warm vs. very cold simulations of Siberian summer temperatures (e.g. a one-point-correlation map of the averaged Siberian SLP and temperature with the northern hemisphere SLP and temperature).

**We thank the reviewer for these interesting suggestions.**

**Likely as a consequence of the many differences within the PMIP ensemble, strong relationships such as the ones suggested by the reviewer have not been found. The strongest pattern we could find is a linear correlation between local (Siberian) JJA temperature anomalies and the large-scale stationary wave pattern anomalies (described by 500hPa geopotential height anomalies with the zonal means removed). The fact that this correlation map resembles a classical stationary wave pattern to us is a strong indication of the importance of this mechanism to explain our findings.**

**We agree with the reviewer that the calculations of the significance provide only limited additional information and we have therefore removed them.**

Regarding the large temperature spread over Eurasia, I was also wondering whether there is a potential link between warm and cold model experiments and the used atmospheric resolution (see below). In any case, the paper would strongly gain from a bit more detailed analysis and discussion of these aspects while the rest of the paper is very well written and does not require notable changes with exception of clarifying the sections about the role of the thermal low.

Specific comments:

Title of the paper: Maybe be more specific and write "glacial summer temperatures"?

**Thanks for the suggestion. We have changed the title accordingly.**

Page 2, line 2: Due to the quite shallow Arctic shelf, sea-level changes during the glacial lead to quite large changes in additionally exposed land during low level stands along the Arctic and Siberian coast. During summer, the additional landmass clearly increases the area which can heat up strongly during boreal summers with 24 hours of daylight. I could imagine that such an effect would be higher in models with a high horizontal resolution. It would be very interesting if you could add some information in the manuscript about individual ensemble members if there are indications that their differences in atmospheric resolution lead to systematic differences in Siberian temperatures.

In this context, there is one recent example where a very coarse resolution simulation has been repeated with the same ocean state and external forcing but using a 4x higher atmospheric resolution with CESM1 (Schenk et al. 2018) for the late glacial. In their supplementary figure 4, they show that a much higher atmospheric resolution with CESM1 predicts considerably warmer summers during the Younger Dryas stadial over Eurasia and Siberia compared to the coarse resolution simulation with CCSM3 despite using the same ocean state. They argue that atmospheric blocking in response to the Fennoscandian Ice Sheet (among other reasons) leads to warmer Eurasian summers. They show that the blocking and hence warmer summers are only captured at high resolution. Is this also the case for the warmest vs. coldest PMIP members?

Given the very strong difference in simulated summer temperatures at a different model resolution by Schenk et al. (2018) and the very important results of other studies concerning the atmospheric flow disturbance by ice sheets (as already cited by the authors on page 3), I would suggest to add a

paragraph about whether atmospheric resolution differences in the presence of large continental ice sheets can partly explain the spread of warming or cooling over Siberia.

**The notion of a resolution dependency of the Siberian LGM temperatures is an interesting one. We have now added the LGM JJA temperature anomalies for the Siberian target region in a table to ease such an analysis. However, when comparing these temperature anomalies with the spatial resolution of the atmospheric models (ranking both and plotting them against each other) we do not find any relationship, not even a hint of it. We have added some text to the concluding section of the manuscript to discuss the matter "Recently, Schenk et al. (2018) showed that the spatial resolution of the atmospheric model is key to obtaining realistic glacial temperature anomalies. However, we do not find any correlation between atmospheric model resolution and Siberian JJA LGM temperature anomalies (Table 1), despite having some models with a resolution very similar to one used by Schenk et al. (2018). We note, however, that we did not perform a dedicated experiment changing only the spatial resolution while keeping all other factors the same."**

Regarding the exposed Arctic shelf during stadials: Is there any geological evidence that glaciations in Siberia might correlate with periods of higher sea-level stands (less exposed Arctic shelf and possibly cooler summers with a weaker thermal low and less advection)?
**We have not been able to find any such information in the geological record.**

Page 2, line 20: Can you give an example which one is good and possibly why?
**This is not easy to do. First of all, some of these studies specifically included new mechanisms in order to obtain a good match (be it for the right reasons or not), and other studies show results of ice sheets models driven by multiple climate models and as a result they obtain very different configurations of the Siberian ice cover. For the introduction part of the current manuscript we don't think it is needed to go into the specifics of all these studies.**

Page 5, line 1: Components of GLAC-1D have been published in different papers. Please add here the reference of the complete version which is Ivanovic et al. (2016).
**Thank you for pointing this out. We have updated the text accordingly.**

Page 5, line 3: Figure 5A is too small to see the important differences in ice sheet heights.
**The details of the differences between these two LGM ice sheet reconstructions are not the focus of this study. In some regions one is higher, in other regions the other reconstruction, a complex picture (as shortly described in the method section) and as such one cannot easily make a connection with large-scale circulation changes. In previous publications on the topic this was often possible because they performed sensitivity studies in which they altered the height of the ice sheets in a controlled maner, or removed one of the ice sheets completely. We have added a comment to the method section referring the readers to the work by Kageyama et al. (2017) for more details.**

Page 5, line 27: The green contour line is not visible. Please add in addition the coordinates for the target region in the manuscript (for the analysed 1 ◦ x1 ◦ grid). Version
**We have improved the readability of the green contour. Giving the coordinates is not feasible since the region is not a rectangle.**

Page 6, lines 13-14: Regarding ". . .could be a consequence of local temperature changes. . .": This is quite certain as the low pressure over Siberia during summer  is a thermal low and not a dynamic low. The sentence should be modified accordingly.
**Indeed in this region warm summer temperatures lead to a low pressure system, a so-called thermal low. However, that is not our point here. Previous work has suggested that local increases in sea-level pressure, driven by large-scale atmospheric circulation changes, lead to**

**a decrease in cloud cover and a resulting increase in surface temperatures. The finding of a negative relationship in our study between local summer temperatures and sea-level pressure rather than a positive one, suggest that this mechanism can't explain the PMIP results. Rather, temperature changes lead to local changes in sea-level pressure (in line with the formation of the thermal low mention by the reviewer). However, we think it is not necessary, and indeed only complicates matters, to describe the background climate characteristics. It is the changes and the sign of possible feedbacks that we are interested in here.**

Page 6, lines 15-16: The link to the Asian monsoon region and possibly other large-scale teleconnections are very important and should be explored a bit more in the manuscript.

**Even though we agree that these are interesting topics, they are really outside of the scope of this manuscript and we have no reason to assume that they are central to the description of the mechanisms driving the large inter-model differences in Siberian summer temperatures.**

Page 6, lines 17-25: The paragraph should be clarified with respect to the low being a thermal low. It appears odd to argue here that a deepening of the low-pressure cell over central Asia (it is not really a cell but rather a diffuse area) should control the amount of warming in Siberia when the deepening of the low is driven by the warming. This might be rather a positive feedback where warming increases convection which lowers the pressure which increases horizontal advection. This implies that another process causes the warming and the change in SLP is only a feedback. Please rewrite accordingly.

**Using geopotential height anomaly maps we now show in an updated figure 3 that local Siberian summer temperatures are linearly correlated with a change in the large-scale circumarctic stationary wave pattern. Far-field surface pressure anomalies are no longer discussed.**

Page 10, lines 21-22: It would be interesting to get a number for the overall temperature change of the northern hemisphere in response to using a different ice sheet in CESM.

**Thanks for pointing this out. We have now added a line "On a large scale, using the GLAC-1D ice-sheet reconstruction leads to a smaller LGM JJA temperature anomaly in the Northern Hemisphere (-6.4∘ C) than the simulation that includes the ICE-6G ice-sheet reconstruction (-7.2∘ C)."**

Page 10, lines 24-25: This again is due to the thermal low which has to deepen with increasing temperature due to an increase in the rise of warm air.

**Since we no longer focus on far-field sea-level pressure changes we assume this issue is resolved.**

Page 10, line 31: The similarity of the spatial anomaly pattern for temperature and SLP can be expected for the behaviour of the thermal low in summer. There has to be another reason for the warming first and the SLP change cannot be the mechanism but rather a positive feedback.

**Since we no longer focus on far-field sea-level pressure changes we assume this issue is resolved.**

Page 15, line 17: Please add a concluding paragraph about which model configuration for CESM (and e.g. which ice sheet) would be plausible for the LGM (no glaciation in Siberia) and why. In this context, can you give some examples about which PMIP models would be plausible for the LGM and absence of Siberian glaciation and which not and why?

**We think this is really not the point of our manuscript, and perhaps in fact inappropriate. Indeed the simulated temperature fields over Siberia (and as a result snow cover and potential ice sheet cover) are very different between models and more in line with geological data in some of them. We show that large changes in simulated temperatures can have many causes,**

**from boundary conditions (ice sheets), to feedbacks (vegetation) to model formulation (atmospheric model). A 'good' simulation can thus result from various combinations of these factors. Moreover, there are indications that during previous glacial periods the ice sheets in northeastern Siberia were more extensive, so a 'good' model should also be able to simulate such a situation. We have added a short comment on the possible implications of our findings for the presence and absence of this ice sheet during various glacial periods.**

Figure 1: The green contour in panel B is not visible.
**Thanks for pointing this out. We have updated it.**

Figure 2: Please strongly increase the size of numbers in the figure as well as the axis description.
**Thanks for pointing this out. We have updated the axis description. However, the size of the numbers in the plots cannot be increased because they will start to overlap and make the figure more difficult to read.**

Figure 3: Are the significant areas really statistically significant globally or only by chance? Given that the correlations may rather represent the thermal low, I'm not sure how this figure helps to understand the spatial spread over Siberia. Pressure gradients and teleconnections might be more suitable as they would represent how the changes of the thermal low interact with remote regions.
**Please refer to the replies given earlier in this rebuttal.**

Figure 8: The red and blue for CAM4/5 is very difficult to see.
**Thanks for pointing this out. We have updated it.**

Table 1: It would be important to add a column here with the temperature difference LGM minus PI over Siberia for each model simulation to identify which models are unusually warm/cold. This would make it easy for others to further explore why which models differ from others. In this way, a potential dependency on the model resolution could be easily identified.
**Thanks for pointing this out. We have included the information to table 1.**

Table 2: Also here the temperature difference LGM minus PI over Siberia would be interesting.
**Thanks for pointing this out. We have included the information to table 2.**

Additional references:
Ivanovic, R. F. et al. Transient climate simulations of the deglaciation 21—9 thousand years before present (version 1)—PMIP4 core experiment design and boundary conditions. Geosci. Model Dev. 9, 2563–2587 (2016).
Schenk, F. et al. Warm summers during the Younger Dryas cold reversal. Nature Commun. 9:1634 (2018).

---

## Author Response (AR1)

Bakker et al. look to understand the mechanisms responsible for Siberian climate at the LGM. To do so, they use a combination of PMIP2/3 simulations and CESM1 sensitivity tests. The authors find that the Siberian region has a large temperature and precipitation spread among models. Using their CESM1 sensitivity tests, Bakker et al. explore the sensitivity of the Siberian region to model physics, ice sheet configuration, and vegetation response. They find that the Siberian temperature response is most significantly influenced by the vegetation, especially when using CAM5, but ice sheet geometry and model physics are also important. Overall, this is a nice study that I believe will be a valuable contribution to understanding climate in a largely overlooked region at the LGM. However, I have a few questions about the model configurations and would like a bit more detailed exploration of mechanisms before publication.

**We thank the reviewer for the kind words and for having a critical look at the manuscript.**

Major Comments:

Additional information about the model setup is required. How was the original LGM simulation, from which these experiments were branched, configured? This is important, because as the authors find, the climate produced by CAM4 and CAM5 can be quite different. Therefore, despite branching from a previous run, I am not convinced that 200 years of spin up is sufficient. Including top-of-atmosphere energy imbalance would provide a first order estimate of how close these simulations are to equilibrium.

**The LGM simulation from which we branched of was run with CAM5 and not including CN-dynamics (Carbon-Nitrogen-Dynamics). This simulation was run for a long time (>1000 years) and was very close to equilibrium, also shown by the TOA imbalance of -0.023Wm$^{-2}$. The relatively short, 200 year, LGM simulation with different model setup resulted in somewhat larger TOA imbalances for the simulations including CN-dynamics (-0.1Wm$^{-2}$ for using CAM4 and -0.185Wm$^{-2}$ using CAM5), however, we deem them sufficiently small, especially considering that the TOA imbalance for the corresponding PI simulations is of similar magnitude (-0.106Wm$^{-2}$ for using CAM4 and -0.117Wm$^{-2}$ using CAM5). The TOA imbalance resulting from the switch from CAM5 to CAM4 (without CN-dynamics) is negligible (0.014 for using CAM4 and -0.023Wm$^{-2}$ using CAM5).**

Also, are 30 year averages enough to produce true climatologies in this region? There are a lot of decadal oscillations that can impact climate for long periods (e.g. Deser et al. 2012). I don't think that this will significantly change results, but I do recommend a quick comparison with a longer average, such as 50 years, to make sure.

**As suggested by the reviewer we compared 30 year and 50 year averages for a limited number of variables and simulations. Indeed the resulting changes in climatology are small, however the relative impact depends on the size of the signal that we are after. For the simulations with the largest LGM to PI JJA temperature difference in the Siberian target region (LGM_CAM5_Veg) the changes in LGM values resulting from the choice of averaging period (30 or 50 years) compared to the LGM to PI anomaly, are 2.1% for JJA surface air temperature and 2.5% for JJA precipitation, with absolute differences of 0.43K for temperature and 4.2mm/year for precipitation. For the simulations with the smallest LGM to PI JJA temperature difference in the Siberian target region (LGM_CAM4_noVeg) the impact of the choice of averaging period relative to the difference in LGM values are 21.5% and 8.3% for respectively JJA temperatures and precipitation in the Siberian target region.**
**We acknowledge the issue raised by the reviewer, but since we focus in the manuscript on the large inter-model differences, we deem the effect sufficiently small.**

Finally, how were the CLM4 cases with "interactive vegetation" spun up? If not spun-up properly, it can take hundreds of years for the carbon cycle to come into equilibrium, which could impact your vegetation distribution.

**The simulation with CN-dynamics were spun up using pre-industrial values. For the regions that have become land under LGM sea-level fall we used a nearest neighbor approach to obtain initial conditions. Indeed, especially the soil carbon pool takes centuries to equilibrate and therefore there are still trends in the different CN-pools. However, the trends in the local (Siberian) vegetation carbon pool, the most relevant for our analysis, is less than 2% of the total PI-to-LGM change in carbon pool over a 50-year period (both in CAM4 and CAM5). We deem this relatively small, but agree that a statement should be included in the manuscript to mention these trends. We added the following to the methodology section "Carbon pools in the litter and soils take centuries to equilibrate. However, we find that the trends are sufficiently small after 200 years to perform a robust analysis of the surface climate. Changes in Siberian (global) vegetation carbon pools are less than 2% (0.6%) of the total PI-to-LGM change) for the model years 150-200"**

Limiting the analyses to JJA limits the mechanistic understanding. Are you sure that the summer changes are mainly a result of summer processes? Also, a more rigorous exploration of the local radiative effects versus heat transport would be useful. For example, albedo and cloud radiative forcings would be more insightful than snow and cloud cover.

**Thank you for this interesting remark. We have performed an additional analysis looking at the seasonal cycle of PMIP multi-model variability for Siberian temperatures, cloud cover and snow cover. For all three variables it is clear that the large increase in differences between the various PMIP models going from PI to LGM is a summer feature. In the other seasons, temperature variance also increases somewhat, but cloud cover variance doesn't change while snow cover variance is in fact decreased.**

**In the updated manuscript we include these figures in the supplement as Figure A1 and mention them in the main text.**

The authors argue for the necessity of additional CESM simulations based in part on the number of variables available for analysis from the PMIP simulations, but proceed to explore only basic outputs from their CESM experiments. Additional analyses to explore why the temperature changes in CESM with different configurations is warranted. At a minimum, areas of perennial snow cover are worth including. What about sea ice? Maybe a PDD and/or energy balance calculation would be insightful. With additional information, the authors could make a much more significant statement about which simulations would produce an ice sheet in Siberia at the LGM. From there, additional model assessment with proxies is possible. Are the models that produce a Siberian ice sheet too cold (probably) or too wet, etc. . .? What does this suggest about Siberian climate at the LGM?

**We agree with the reviewer that many more interesting analyses could be performed using the set of CESM simulations. However, we want to stress that our main reason to include the CESM results is to be able to show which differences between PMIP simulations can potentially lead to large differences in Siberian JJA temperatures (ice sheets, atmopheric model and vegetation feedback). Such a separation of factors is not possible for the PMIP ensemble.**

**More CESM results for the different PI and LGM simulations are now provided in a new table (Table 3), including Siberian temperatures, minimum snow cover, cloud cover, precipitation and sea-level pressure.**

Specific:
P1 Line 20: Further south than 50 ◦ N in many locations in North America.
**We have changed the line to read "down to ~40 ◦ N in some areas.**

P1 Line 21: Much of Alaska also did not have ice.
**Indeed much of Alaska was also ice free during the LGM. We've changed the line to read "A notable exception was..."**

P1 Line 25: Didn't some of these modeling studies limit their ice domain to exclude Siberia? Double check.
**The work by Abe-Ouchi et al. (2013) was a free running modelling experiment that did not exclude Siberia from their domain (in fact they do simulate a Siberian ice sheet when applying an additional cooling factor). The other two studies are combined model-data driven reconstructions and as such they use the absence of an ice sheet in reconstructions as target in their modelling exercise.**

P2 Line 2: Citation for the sea level statement?
**We deem the notion that sea level was globally lower during the LGM as common knowledge and as such a reference is not needed here.**

P2 Line 30: This dust feedback is mentioned in earlier (e.g. Mahowald et al., 1999; Ganopolski et al., 2010). What about the direct radiative effect of dust (e.g. Schneider et al., 2006)?
**Thanks for pointing this out. We have added a reference to Mahowald et al. 1999 on line 27. We did not add it at line 30 as those are studies specifically discuss the evolution of the Siberian ice sheet in relation with LGM dust deposition. We prefer not to include a reference to Schneider et al. (2006) since they do not specifically discuss Siberia.**

P4 Line 1: Link is messed up.
**Thank you for pointing this out. I has been corrected.**

P4 Line 10: Should be 1.9x2.5。
**Thank you for pointing that out. We have adjust it.**

P5 Line 16: Shouldn't this citation be for an ice sheet reconstruction paper? Peltier et al. (2015) maybe?
**This reference has been updated to Ivanovic et al. (2016).**

P5 Line 34: Not sure that ensemble is the correct word.
**We have changed our wording. When referring to a small set of CESM experiments we call it a 'set of experiments', and only when we discuss all the CESM simulations combined do we now refer to it as an 'ensemble'.**

P6 Line 5: Need to spell out LGM_CAM5_noVeg first.
**Thank you for pointing this out. We have removed the acronym.**

P6 Line 14: Why not look at the snow cover in the model?
**We agree that this line is confusing and have therefor removed it. In the manuscript we do look at snow cover.**

P6 Line 4: Cloud radiative forcing would be more insightful.
**Cloud radiative forcing is unfortunately not available for all PMIP2 and PMIP3 LGM simulations.**

P6 Line 13: Did you analyze CCSM3, as used in Liakka et al. (2016), to better understand this discrepancy? Could use a bit of additional discussion.

**We do not think that there is a discrepancy per se. Possibly some of the models in the PMIP ensemble show results that resemble the one described by Liakka et al (2016). What we argue is that in the PMIP ensemble as a whole this mechanism does not seem to be the leading explanation for the temperature changes in Siberia. We have updated the main text accordingly.**

Figure 1: Make the continental outlines thicker.
**Figure has been updated**

Figure 2: Darker green would make it easier to see.
**Thank you for the suggestion. We improved the clarity of the figure by adding a magnified portion of the map highlighting the "target region".**

Figure 3: Add winds and/or height anomalies to better highlight the circulation changes.
**The contents of this figure have been changed from sea-level pressure to geopotential height anomalies at 500hPa to provide a much more direct indication of large-scale atmospheric circulation changes.**

P10 Line 16: Why not plot the same variables as in the PMIP runs with CESM?
**By showing geopotential height anomalies at 500hPa in figure 3 for the PMIP models and by adding summary information on CESM-based Siberian temperatures, minimum snow cover, cloud cover, precipitation and sea-level pressure in a new table (Table 3), we now effectively show the same variables for PMIP and CESM results as long as they are available.**

P10 Line 22: How is surface roughness over the ice sheets configured? The results of Brady et al. (2013) suggest that this is important.
**This is indeed one of those things that are uncertain for LGM simulations. We have chosen a simplified approach assigning a constant value similar to other areas that are ice covered at present day, but we agree that this is yet another mechanism that could impact temperatures since the sensitivity in the northeast Siberia to perturbations of the large-scale circulation is so large.**

P10 Line 8-2?: It would be great to plot some of the differences mentioned.
**We are not entirely sure what the reviewer is referring to in this comment, but assuming it is on the differences between CAM4 and CAM5, we would argue that such an analysis should really be performed by the experts who know all the details of the two atmospheric models.**

P11 Line 20: How do you define vegetation density?
**We use the term density here to describe in general how much vegetation there is per unit area, which in the model is mainly determined by the combination of the leaf area index and the stem area index.**

P11 Line 22: This vegetation feedback has been found to be important for Arctic climate before (e.g. Jahn et al., 2005; Tabor et al., 2014).
**Thanks for pointing this out. On page 11 we have added a line acknowledging this "Previous studies also found an important role of vegetation feedbacks in defining LGM Arctic temperatures (Jahn et al., 2005)."**

Figure 5 A: There must be a strong local feedback in Siberia. Maybe plot snow cover or albedo?
**Indeed there are multiple strong local feedbacks (snow cover, cloud cover changes etc). In a new table we have included information for the various CESM simulations on temperature, precipitation, cloud cover, snow cover and sea-level pressure in the Siberian target region.**

**Indeed temperature and the summer snow fraction are related, showing a local feedback. We added text to the main text describing some of the features of the data in this new table.**

P11 Line 23: Does this mean the vegetation dies?

**The different vegetation zones move southward, including the zone that has very little vegetation cover. In CESM the plant functional types are prescribed and thus not changing between the different experiments. They prescribe a mixture of different PFT's in every grid cell and the apparent southward shift of the vegetation zones is thus a change in the dominance of certain PFT's within the individual grid cells. We currently can't tell how the results would change if the PFT's would be interactively calculated using a full dynamic vegetation model.**

P11 Line 26: Does Lawrence et al. (2011) discuss this Arctic LAI issue?

**Thanks for pointing this out. Indeed Lawrence et al. (2011) also discuss that in CLM4 Siberian surface and soil temperatures are biased low (compared to observations) while CLM3 they were biased somewhat high. We added a short statement in the manuscript "….and is in line with the cold bias in modelled Siberian surface temperatures described by Lawrence et al. (2011)."**

Figure 6: How were your PI runs configured for your LGM-PI anomalies? The vectors are very hard to see in panel C. Please change the color.

**We do not understand the first part of this question. What is meant with configured in this context? We have made the vectors more clear.**

Figure 7: Extend the temperature range in panel B.

**Thanks for pointing this out. We have updated the figure to be more readable.**
Geological evidence has shown that Siberia was partially glaciated during some glacial states while it kept mostly ice-free during others. Different previous studies have explored several potential explanations for these differences but a consensus is still lacking. Bakker et al. show that the ensemble of climate model experiments from PMIP2 and PMIP3 shows a very large spread in their simulated glacial summer (JJA) temperatures for the last glacial maximum (LGM) over Siberia. Bakker et al. argue that the large model spread could be an indication for a real "hypersensitivity" of glacial summer temperatures over Siberia, and hence regional glaciation itself. To explore some of the possible factors which may result in climatic differences over Siberia, they conduct several sensitivity simulations with CESM and show that the spread in simulations resulting from different ice sheet heights, vegetation feedback or changes in atmospheric physics of CAM4/5 can cause an equally large spread ($\sim$20 K) as the PMIP model ensemble ($\sim$24 K).

Overall, the manuscript is very well written and provides interesting insights into the problem of glacial summer temperature hypersensitivity and how it might explain the absence or presence of glaciation in Siberia during different glacials. However, the potential reasons for what may cause the large simulated temperature spread over Siberia could be explored in a bit more detail. I generally recommend publication in Climate of the Past after adding some more analysis to explain the summer temperature discrepancies.
**We thank the reviewer for the kind words and for having a critical look at the manuscript.**

General comments:
The study is very well written and presents very interesting and important aspects to better understand the possibly real "hypersensitivity" of the Siberian climate during glacials as well as the behaviour of models. Regarding the analysed variables in the manuscript, it is a bit difficult to understand whether local radiative processes (e.g. what about albedo, spring snow cover and lagged warming?) or large-scale temperature advection play a major role for the temperature spread – or both. Because Siberia builds up a spatially widespread thermal low during summer, the correlation between summer temperature and SLP can be expected to be mainly temperature driven. Increasing temperature will hence cause lower SLP which then can increase horizontal advection into Siberia. Consequently, changes in SLP would be rather a feedback to the warming (or cooling) and not the mechanism which causes the effect.
**We agree with the reviewer that it is difficult to disentangle local versus large-scale effects on Siberian temperatures. This is especially true when doing so through the analysis of sea-level pressure fields. Fortunately we have found out that for all but one PMIP2/3 LGM simulation also geopotential height fields are available, and this makes for a more direct line of arguments and, in our opinion, a more convincing analysis to show that changes in the large-scale, circumarctic atmospheric circulation are indeed the cause of the spread in simulated Siberian JJA temperatures.**
**In the updated manuscript we show PMIP results for geopotential height anomalies at 500hPa (with the zonal mean removed), and together with the existing CESM geopotential height results we argue that both clearly show large-scale anomaly patterns that resemble a classical stationary wave pattern (wave number 2) and therefore indicate changes in the large-scale atmospheric circulation. Moreover, the four center of action in the PMIP based figure (figure 3) are the regions for which the relationship is significant, strengthening in our view the link with large-scale circulation as a driver. Based on CESM results we have already shown that as a result of these circulation changes, meridional heat transport into the region under discussion increases. This then reinforced the climatological thermal low in Siberia.**

I also wonder whether the correlation in Fig. 3 is really statistically significant in terms of field significance given the low spatial degrees of freedom of SLP and that the relatively small regions with statistically significant correlations might be just those which are allowed to be significant by chance. In general, I would rather expect that the large-scale gradients in the pressure and temperature field e.g. relative to the Arctic and Tropics are important for temperature advection into Siberia. It would be interesting to see some analysis of the large-scale wind fields or pressure gradients and how different they are with respect to the model spread e.g. of the warmest vs. coldest PMIP member. I was also wondering if large-scale teleconnections might be very different for very warm vs. very cold simulations of Siberian summer temperatures (e.g. a one-point-correlation map of the averaged Siberian SLP and temperature with the northern hemisphere SLP and temperature).

**We thank the reviewer for these interesting suggestions.**

**Likely as a consequence of the many differences within the PMIP ensemble, strong relationships such as the ones suggested by the reviewer have not been found. The strongest pattern we could find is a linear correlation between local (Siberian) JJA temperature anomalies and the large-scale stationary wave pattern anomalies (described by 500hPa geopotential height anomalies with the zonal means removed). The fact that this correlation map resembles a classical stationary wave pattern and that the four main centers of action are the regions for which the correlations are significant are to us strong indications of the importance of this mechanism to explain our findings. It is in this light that we would prefer to keep the significance levels in the figure. We have highlighted the latter points in the updated manuscript.**

Regarding the large temperature spread over Eurasia, I was also wondering whether there is a potential link between warm and cold model experiments and the used atmospheric resolution (see below). In any case, the paper would strongly gain from a bit more detailed analysis and discussion of these aspects while the rest of the paper is very well written and does not require notable changes with exception of clarifying the sections about the role of the thermal low.

Specific comments:
Title of the paper: Maybe be more specific and write "glacial summer temperatures"?

**Thanks for the suggestion. We have changed the title accordingly.**

Page 2, line 2: Due to the quite shallow Arctic shelf, sea-level changes during the glacial lead to quite large changes in additionally exposed land during low level stands along the Arctic and Siberian coast. During summer, the additional landmass clearly increases the area which can heat up strongly during boreal summers with 24 hours of daylight. I could imagine that such an effect would be higher in models with a high horizontal resolution. It would be very interesting if you could add some information in the manuscript about individual ensemble members if there are indications that their differences in atmospheric resolution lead to systematic differences in Siberian temperatures.

In this context, there is one recent example where a very coarse resolution simulation has been repeated with the same ocean state and external forcing but using a 4x higher atmospheric resolution with CESM1 (Schenk et al. 2018) for the late glacial. In their supplementary figure 4, they show that a much higher atmospheric resolution with CESM1 predicts considerably warmer summers during the Younger Dryas stadial over Eurasia and Siberia compared to the coarse resolution simulation with CCSM3 despite using the same ocean state. They argue that atmospheric blocking in response to the Fennoscandian Ice Sheet (among other reasons) leads to warmer Eurasian summers. They show that the blocking and hence warmer summers are only captured at high resolution. Is this also the case for the warmest vs. coldest PMIP members?

Given the very strong difference in simulated summer temperatures at a different model resolution by Schenk et al. (2018) and the very important results of other studies concerning the atmospheric

flow disturbance by ice sheets (as already cited by the authors on page 3), I would suggest to add a paragraph about whether atmospheric resolution differences in the presence of large continental ice sheets can partly explain the spread of warming or cooling over Siberia.

**The notion of a resolution dependency of the Siberian LGM temperatures is an interesting one. We have now added the PMIP-based LGM JJA temperature anomalies for the Siberian target region in table 1 to allow for such an analysis. When comparing these temperature anomalies with the spatial resolution of the atmospheric models (ranking both and plotting them against each other) we do not find any relationship. We have added a short comment to the concluding section of the manuscript to discuss the matter "Recently, Schenk et al. (2018) showed that the spatial resolution of the atmospheric model is key to obtaining realistic glacial temperature anomalies. However, we do not find any correlation between atmospheric model resolution and Siberian JJA LGM temperature anomalies (Table 1), despite having some models with a resolution very similar to one used by Schenk et al. (2018). We note, however, that we did not perform a dedicated experiment changing only the spatial resolution while keeping all other factors the same."**

Regarding the exposed Arctic shelf during stadials: Is there any geological evidence that glaciations in Siberia might correlate with periods of higher sea-level stands (less exposed Arctic shelf and possibly cooler summers with a weaker thermal low and less advection)?

**Perhaps MIS6 could represent such a geological period, but if such a correlation exists we dare not say based on the little data that we have.**

Page 2, line 20: Can you give an example which one is good and possibly why?

**This is not easy to do. First of all, some of these studies specifically included new mechanisms in order to obtain a good match (be it for the right reasons or not), and other studies show results of ice sheet models driven by multiple climate models and as a result they obtain very different configurations of the Siberian ice cover. For the introduction part of the current manuscript we don't think it is needed to go into the specifics of all these studies.**

Page 5, line 1: Components of GLAC-1D have been published in different papers. Please add here the reference of the complete version which is Ivanovic et al. (2016).

**Thank you for pointing this out. We have updated the text accordingly.**

Page 5, line 3: Figure 5A is too small to see the important differences in ice sheet heights.

**We agree that it is not easy from figure 5A to see the details of the differences in ice sheet height. An additional figure only showing these differences could be added to the supplementary material. However, we would like to stress that the details of the differences between these two LGM ice sheet reconstructions are not the focus of this study. In previous dedicated studies sensitivity experiments were performed in which they altered the height of the ice sheets, or even an individual ice sheet, in a controlled manner, or even removed one of the ice sheets completely. That allows for an in-depth study of its impact. When comparing the two ice sheet reconstructions used here we see that in some regions the one ice sheet reconstruction is higher, in other regions the other reconstruction, a complex picture (as shortly described in the method section) and as such one cannot easily make a connection with large-scale circulation changes. For the reader that is interested in the details of the two reconstructions we propose to add a reference to the work by Kageyama et al. (2017).**

Page 5, line 27: The green contour line is not visible. Please add in addition the coordinates for the target region in the manuscript (for the analysed 1 ∘ x1 ∘ grid). Version

**We improved the clarity of the figure by adding a magnified portion of the map highlighting the "target region". Giving the exact coordinates is not feasible since the region is not a**

**rectangle, but we have added a note about its approximate location "roughly located between 120E-180E and 70N-75N".**

Page 6, lines 13-14: Regarding ". . .could be a consequence of local temperature changes. . .": This is quite certain as the low pressure over Siberia during summer is a thermal low and not a dynamic low. The sentence should be modified accordingly.

**We agree with the reviewer that this section is not clear enough.**

**Indeed in this region warm summer temperatures lead to a low pressure system, a so-called thermal low. Previous work has suggested that local increases in sea-level pressure, driven by large-scale atmospheric circulation changes, lead to a decrease in cloud cover and a resulting increase in surface temperatures. The finding of a negative relationship in our study between local summer temperatures and sea-level pressure rather than a positive one, suggest that this mechanism can't explain the majority of PMIP results. Instead, in line with the formation of the thermal low mention by the reviewer, temperature changes lead to local changes in sea-level pressure.**

**To bring this message across to the reader this section now reads: "Moreover, a strong anticorrelation is found in the PMIP LGM simulations between JJA temperature and sea-level pressure anomalies over the Siberian target region (R=-0.72; p<0.05; Figure 2C): a more positive temperature anomaly locally creates a thermal low and hence corresponds to a lower sea-level pressure anomaly. Concurrently, higher sea-level pressure anomalies correspond to more positive cloud cover anomalies (R=0.50; p<0.05; Figure 2D). Liakka et al. (2016) found in their model that higher pressure is associated with lower cloud cover that in turn leads to an increase in JJA temperatures, but our results suggests that this is not the leading mechanism in the majority of PMIP LGM results."**

Page 6, lines 15-16: The link to the Asian monsoon region and possibly other large-scale teleconnections are very important and should be explored a bit more in the manuscript.

**We agree with the reviewer that these are interesting topics. However, we have found no indications that they are central to the description of the mechanisms driving the large inter-model differences in Siberian summer temperatures. As such we would like to keep the current focus of the manuscript.**

Page 6, lines 17-25: The paragraph should be clarified with respect to the low being a thermal low. It appears odd to argue here that a deepening of the low-pressure cell over central Asia (it is not really a cell but rather a diffuse area) should control the amount of warming in Siberia when the deepening of the low is driven by the warming. This might be rather a positive feedback where warming increases convection which lowers the pressure which increases horizontal advection. This implies that another process causes the warming and the change in SLP is only a feedback. Please rewrite accordingly.

**Using geopotential height anomaly maps (updated figure 3), we now show that local Siberian summer temperatures are linearly correlated with a change in the large-scale circumarctic stationary wave pattern. This provides indeed a much more direct link and therefor a discussion of the difficult to interpret 'far-field surface pressure anomalies' is no longer needed.**

Page 10, lines 21-22: It would be interesting to get a number for the overall temperature change of the northern hemisphere in response to using a different ice sheet in CESM.

**Thanks for pointing this out. We have now added a line "On a large scale, using the GLAC-1D ice-sheet reconstruction leads to a smaller LGM JJA temperature anomaly in the Northern Hemisphere (-6.4◦ C) than the simulation that includes the ICE-6G ice-sheet reconstruction (-7.2◦ C)."**

Page 10, lines 24-25: This again is due to the thermal low which has to deepen with increasing temperature due to an increase in the rise of warm air.

Page 10, line 31: The similarity of the spatial anomaly pattern for temperature and SLP can be expected for the behaviour of the thermal low in summer. There has to be another reason for the warming first and the SLP change cannot be the mechanism but rather a positive feedback.

**As mentioned above, we hope that the reviewer agrees that these issues are resolved by focusing on geopotential height anomalies rather than surface pressure anomalies.**

Page 15, line 17: Please add a concluding paragraph about which model configuration for CESM (and e.g. which ice sheet) would be plausible for the LGM (no glaciation in Siberia) and why. In this context, can you give some examples about which PMIP models would be plausible for the LGM and absence of Siberian glaciation and which not and why?

**We do not think this is feasible based on our analysis. The simulated temperature fields over Siberia (and as a result snow cover and potential ice sheet cover) are very different between models and more in line with geological data in some of them. We show that large changes in simulated temperatures can have many causes, from boundary conditions (ice sheets), to feedbacks (vegetation) to model formulation (atmospheric model). A 'good' simulation can thus result from various combinations of these factors. Moreover, there are indications that during previous glacial periods ice sheets existed in northeastern Siberia, so a 'good' model should also be able to simulate such a situation. We have added a short comment to the very end of the manuscript on the possible implications of our findings for the presence and absence of this ice sheet during various glacial periods "The combination of these factors, accompanied by local feedbacks can lead to strongly divergent summer temperatures in the region, which during some glacial periods could be sufficiently low to allow for the buildup of an ice sheet, while during other glacials, above-freezing summer temperatures will prevent a multi-year snow-pack, and hence an ice sheet, from forming. Finally, this high sensitivity of Siberian LGM summer temperatures in different climate models will present a major challenge in future modelling efforts using coupled ice-sheet-climate models."**

Figure 1: The green contour in panel B is not visible.

**Thanks for pointing this out. We have updated it.**

Figure 2: Please strongly increase the size of numbers in the figure as well as the axis description.

**Thanks for pointing this out. We have updated the axis description. However, the size of the numbers in the plots cannot be increased because they will start to overlap and make the figure more difficult to read.**

Figure 3: Are the significant areas really statistically significant globally or only by chance? Given that the correlations may rather represent the thermal low, I'm not sure how this figure helps to understand the spatial spread over Siberia. Pressure gradients and teleconnections might be more suitable as they would represent how the changes of the thermal low interact with remote regions.

**We agree that the correlations are not particularly strong. In the updated figure 3 we are now looking at the temperature relationship with the large-scale stationary wave pattern anomalies (described by 500hPa geopotential height anomalies with the zonal means removed), which provides a much more direct measure of the large-scale atmospheric circulation. The fact that this correlation map resembles stationary wave pattern and that the four main centers of action are the regions for which the correlations are significant, to us are strong indications of the importance of this mechanism to explain our findings. It is in this light that we would prefer to keep the significance levels in the figure. We have highlighted the latter points in the updated manuscript.**

Figure 8: The red and blue for CAM4/5 is very difficult to see.

**Thanks for pointing this out. We have updated it.**

Table 1: It would be important to add a column here with the temperature difference LGM minus PI over Siberia for each model simulation to identify which models are unusually warm/cold. This would make it easy for others to further explore why which models differ from others. In this way, a potential dependency on the model resolution could be easily identified.
**Thanks for pointing this out. We have included the information to table 1.**

Table 2: Also here the temperature difference LGM minus PI over Siberia would be interesting.
**Thanks for pointing this out. We have included the information to table 3.**

Additional references:

[revised manuscript text omitted]

---

## Author Response (AR2)

[revised manuscript text omitted]

Response to reviewer 1 in second round of reviews:

Reviewer comments in **bold**.

**Bakker et al. generally do a good job addressing my previous comments. However, I still have a few suggestions for the revised manuscript that should be considered before publication.**

**I agree that the TOA imbalances are not especially large. But, it is possible that regionally some of the ongoing changes are large. It would be good to mention the TOA imbalances in the text.**

We have added the following to the manuscript: "Top-of-the-atmosphere imbalances in the simulations including the carbon-nitrogen cycle are -0.1\unit{Wm^{-2}} and -0.185\ unit{Wm^{-2}} using the CAM4 and CAM5 atmospheric models, respectively."

**-Any reason not to use 50 year averages? Do you think that the differences between 30 and 50 year averages are the result of internal variability or adjustment to a new equilibrium?**

The only real reason is simply that 50 years of data are not available (anymore) for all simulations. Because we don't think it matters for the analysis if we use 30 or 50 year averages we think that is not a major issue. Except for some of the slower carbon pools, all relevant parts of the climate system are well in equilibrium after more than 100 years of simulation (starting from a previous long LGM simulation) and so we assume that the differences between the 30 and 50 year averages (years 170-200 or 150-200, respectively) are internal variability.

**-Please reword – "less than 2% of the total PI-to-LGM change in carbon pool over a 50-year period". Does this mean less than 2% of the total PI-to-LGM change in carbon pool over a 50-year period in the Siberian region? If so, is there a large change in carbon at the start of the simulation or is there a gradual adjustment? If not, are you comparing with the global carbon change between PI and LGM? This would not be very useful.**

In the manuscript it reads: "Changes in Siberian (global) vegetation carbon pools amount to less than 2\% (0.6\%) of the total PI-to-LGM change for the model years 150-200." So both the Siberian and global values are given. This statement seems quite clear to us. Unfortunately we don't have the model output to investigate the temporal evolution of these carbon changes.

**-Maybe mention the uncertainty in surface roughness in the text.**

The following has been added to the manuscript: "Changes in surface roughness resulting from the ice sheet changes are highly uncertain and have not been taken into account."

**-"They prescribe a mixture of different PFT's in every grid cell and the apparent southward shift of the vegetation zones is thus a change in the dominance of certain PFT's within the individual grid cells." I do not think this statement is correct. The percentage of vegetation of different types does not change within a grid cell but the**

**vegetation can die. The biome composition is prescribed. For example, a grid cell can be 50% shrub and 50% boreal forest. The boreal forest might die. As a result, the grid cell will be 50% shrub and 50% dead boreal forest.**

The reviewer is very right to point this out. However, I don't see this statement in the manuscript. Perhaps the reviewer looked at a previous version of the manuscript? The meridional movements of the tree and shrub lines are based on the leaf area index (that is mentioned in the manuscript), not the dominance of a certain type of PFT.

**-The figures have been improved. I still find the light green very difficult to see. I would recommend changing it.**

We agree that it is not easy to see also because the figures are quite small, but because of the addition of the magnification of the target region we think it is now sufficiently clear for the reader.

Response to reviewer 2 in second round of reviews

Reviewer comments in **bold**.

**The authors have clearly improved their already well-written manuscript. The additional analysis of geopotential height anomalies provides an important previously less clear aspect highlighting how the identified wave train pattern helps to explain the large-scale temperature advection anomalies. I tend to agree with reviewer #1 that it is tempting to ask for an evaluation of several more aspects incl. a dependency on sea-ice, ice sheet (and resolution) configuration impacts in PMIP models on the wave train pattern etc. But this should be rather done in additional studies for which Bakker et al. create an important case to continue from. In this context, their added information on individual PMIP model resolutions and their temperature spreads over Siberia (Table 1) as well as a comparison of own model experiments in Table 3 are very useful to select certain models and variables for follow-up studies.**
**Overall, the authors have adequately addressed all points and I, therefore, suggest publication as is.**

**Technical remarks:**
**Figure 5C and 6C: To avoid red-green colour blindness issues, I would recommend using another colour (e.g. black or dark grey) and try slightly thicker lines. It is still difficult to see the vectors.**

The vectors are now done in dark grey and made a little thicker.

**Table 1 and 3: I'm very grateful to the authors for putting these tables together. They are very interesting and useful to verify the large spread between models in a quantified way. Perhaps use a precision of only one decimal rather than two for the numbers.**

Done